# ProlificDreamer: High-Fidelity and Diverse Text-to-3D Generation with Variational Score Distillation

**Zhengyi Wang**[*1,3], **Cheng Lu**[*1], **Yikai Wang**[1], **Fan Bao**[1,3],
**Chongxuan Li**[† 2], **Hang Su**[1,4], **Jun Zhu**[† 1,3,4]

[1]Dept. of Comp. Sci. & Tech., BNRist Center, Tsinghua-Bosch Joint ML Center,
Tsinghua University; [2]Gaoling School of Artificial Intelligence, Renmin University of China,
Beijing Key Laboratory of Big Data Management and Analysis Methods, Beijing, China
[3]ShengShu, Beijing, China; [4]Pazhou Laboratory (Huangpu), Guangzhou, China
{wang-zy21, bf19}@mails.tsinghua.edu.cn; lucheng.lc15@gmail.com;
yikaiw@outlook.com; chongxuanli@ruc.edu.cn; suhangss@tsinghua.edu.cn
dcszj@tsinghua.edu.cn

## Abstract

Score distillation sampling (SDS) has shown great promise in text-to-3D generation by distilling pretrained large-scale text-to-image diffusion models, but suffers from over-saturation, over-smoothing, and low-diversity problems. In this work, we propose to model the 3D parameter as a random variable instead of a constant as in SDS and present *variational score distillation* (VSD), a principled particle-based variational framework to explain and address the aforementioned issues in text-to-3D generation. We show that SDS is a special case of VSD and leads to poor samples with both small and large CFG weights. In comparison, VSD works well with various CFG weights as ancestral sampling from diffusion models and simultaneously improves the diversity and sample quality with a common CFG weight (i.e., 7.5). We further present various improvements in the design space for text-to-3D such as distillation time schedule and density initialization, which are orthogonal to the distillation algorithm yet not well explored. Our overall approach, dubbed *ProlificDreamer*, can generate high rendering resolution (i.e., $512 \times 512$) and high-fidelity NeRF with rich structure and complex effects (e.g., smoke and drops). Further, initialized from NeRF, meshes fine-tuned by VSD are meticulously detailed and photo-realistic. Project page: https://ml.cs.tsinghua.edu.cn/prolificdreamer/.

## 1 Introduction

3D content and technologies enable us to visualize, comprehend, and interact with complex objects and environments that are reflective of our real-life experiences. Their pivotal role extends across a wide array of domains, encompassing architecture, animation, gaming, and the rapidly evolving fields of virtual and augmented reality. In spite of the extensive applications, the production of premium 3D content often remains a formidable task. It necessitates a significant investment of time and effort, even when undertaken by professional designers. This challenge has prompted the development of text-to-3D methods [19, 31, 34, 20, 4, 29, 55, 16]. By automating the generation of 3D content based on textual descriptions, these innovative methods present a promising way towards streamlining the 3D content creation process. Furthermore, they stand to make this process more accessible, potentially encouraging a significant paradigm shift in the aforementioned fields.

---

[*]Equal contribution; [†] Corresponding authors.

37th Conference on Neural Information Processing Systems (NeurIPS 2023).

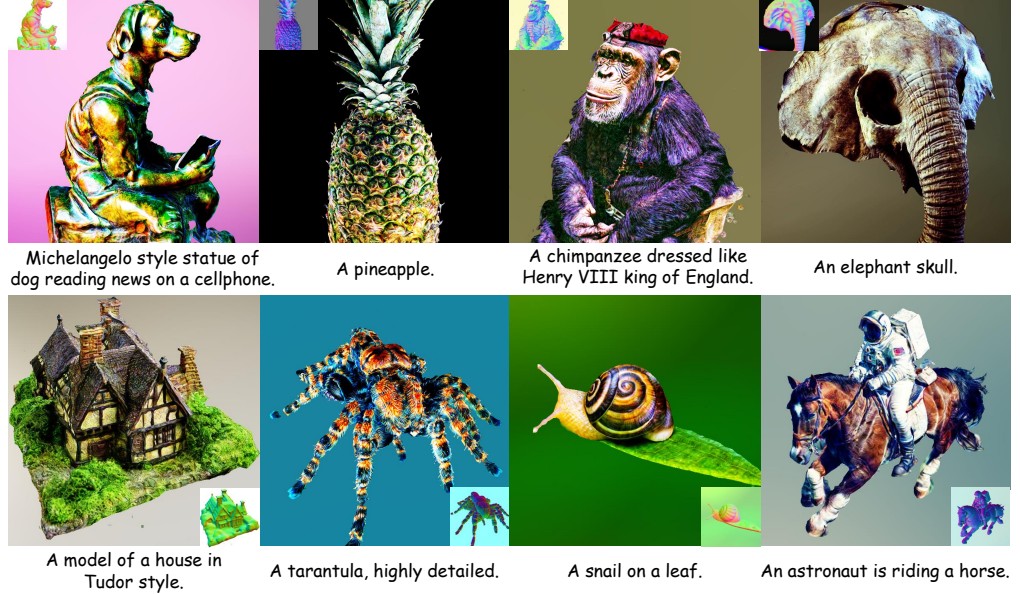

(a) ProlificDreamer can generate meticulously detailed and photo-realistic 3D textured meshes.

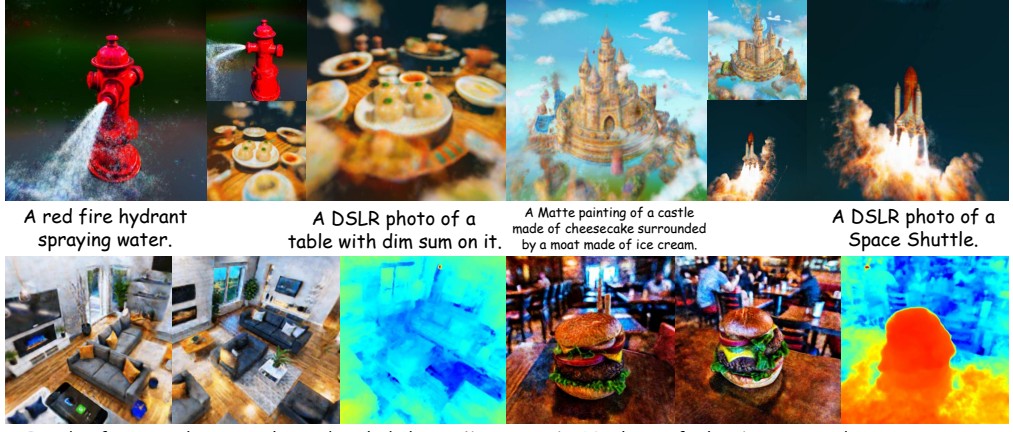

(b) ProlificDreamer can generate high rendering resolution (i.e., $512 \times 512$) and high-fidelity NeRF with rich structures and complex effects. Besides, the bottom results show that ProlificDreamer can generate complex scenes with $360°$ views because of our *scene initialization* (see Sec. 4.2).

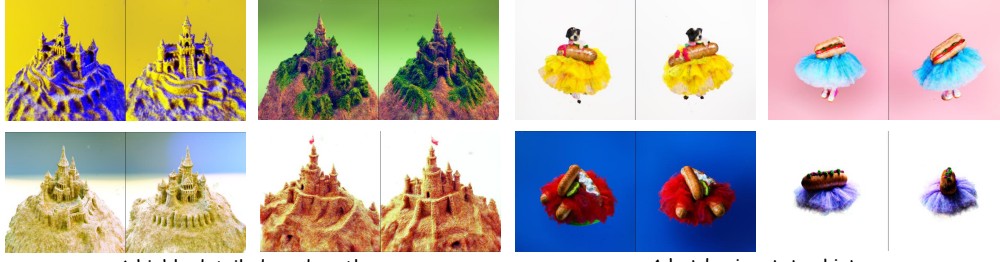

(c) ProlificDreamer can generate diverse and semantically correct 3D scenes given the same text.

Figure 1: Text-to-3D samples generated by ProlificDreamer from scratch. Our base model is Stable Diffusion and we do not employ any other assistant model or user-provided shape guidance (see Table 1). See our accompanying videos in our project page for better visual quality.

Diffusion models [46, 14, 49] have significantly advanced text-to-image synthesis [36, 41, 38, 1], particularly when trained on large-scale datasets [43]. Inspired by these developments, DreamFusion [34] employs a pretrained, large-scale text-to-image diffusion model for the generation of 3D content from text in the wild, circumventing the need for any 3D data. DreamFusion introduces the Score Distillation Sampling (SDS) algorithm to optimize a single 3D representation such that the image rendered from any view maintains a high likelihood as evaluated by the diffusion model, given the text. Despite its wide application [20, 4, 29, 55], empirical observations [34] indicate that SDS often suffers from over-saturation, over-smoothing, and low-diversity problems, which have yet to be thoroughly explained or adequately addressed. Additionally, orthogonal elements in the design space for text-to-3D, such as rendering resolution and distillation time schedule, have not been fully explored, suggesting a significant potential for further improvement. In this paper, we present a systematic study of all these elements to obtain elaborate 3D representations.

We first present *Variational Score Distillation* (VSD), which treats the corresponding 3D scene given a textual prompt as a *random variable* instead of a single point as in SDS [34]. VSD optimizes a distribution of 3D scenes such that the distribution induced on images rendered from all views aligns as closely as possible, in terms of KL divergence, with the one defined by the pretrained 2D diffusion model (see Sec. 3.1). Under this variational formulation, VSD naturally characterizes the phenomenon that multiple 3D scenes can potentially align with one prompt. To solve it efficiently, VSD adopts particle-based variational inference [23, 3, 9], and maintains a set of 3D parameters as particles to represent the 3D distribution. We derive a novel gradient-based update rule for the particles via the Wasserstein gradient flow (see Sec. 3.2) and guarantee that the particles will be samples from the desired distribution when the optimization converges (see Theorem 2). Our update requires estimating the score function of the distribution on diffused rendered images, which can be efficiently and effectively implemented by a low-rank adaptation (LoRA) [18, 40] of the pretrained diffusion model. The final algorithm alternatively updates the particles and score function.

We show that SDS is a special case of VSD, by using a single-point Dirac distribution as the variational distribution (see Sec. 3.3). This insight explains the restricted diversity and fidelity of the generated 3D scenes by SDS. Moreover, even with a single particle, VSD can learn a parametric score model, potentially offering superior generalization over SDS. We also empirically compare SDS and VSD in 2D space by using an identity rendering function that isolates other 3D factors. Similar to ancestral sampling from diffusion models, VSD is able to produce realistic samples using a normal CFG weight (i.e., 7.5). In contrast, SDS exhibits inferior results, sharing the same issues previously observed in text-to-3D, such as over-saturation and over-smoothing [34].

We further systematically study other elements orthogonal to the algorithm for text-to-3D and present a clear design space in Sec. 4. Specifically, we propose a high rendering resolution of $512 \times 512$ during training and an annealed distilling time schedule to improve the visual quality. We also propose *scene initialization*, which is crucial for complex scene generation. Comprehensive ablations in Sec. 5 demonstrate the effectiveness of all the aforementioned elements particularly for VSD. Our overall approach can generate high-fidelity and diverse 3D results. We term it as *ProlificDreamer*[2].

As shown in Fig. 1 and Sec. 5, ProlificDreamer can generate $512 \times 512$ rendering resolution and high-fidelity Neural Radiance Fields (NeRF) with rich structure and complex effects (e.g., smoke and drops). Besides, for the first time, ProlificDreamer can successfully construct complex scenes with multiple objects in $360°$ views given the textual prompt. Further, initialized from the generated NeRF, ProlificDreamer can generate meticulously detailed and photo-realistic 3D textured meshes.

## 2  Background

We present preliminaries on diffusion models, score distillation sampling, and 3D representations.

**Diffusion models.** A diffusion model [46, 14, 49] involves a forward process $\{q_t\}_{t \in [0,1]}$ to gradually add noise to a data point $\boldsymbol{x}_0 \sim q_0(\boldsymbol{x}_0)$ and a reverse process $\{p_t\}_{t \in [0,1]}$ to denoise/generate data. The forward process is defined by $q_t(\boldsymbol{x}_t|\boldsymbol{x}_0) := \mathcal{N}(\alpha_t \boldsymbol{x}_0, \sigma_t^2 \boldsymbol{I})$ and $q_t(\boldsymbol{x}_t) := \int q_t(\boldsymbol{x}_t|\boldsymbol{x}_0) q_0(\boldsymbol{x}_0) \mathrm{d}\boldsymbol{x}_0$, where $\alpha_t, \sigma_t > 0$ are hyperparameters satisfying $\alpha_0 \approx 1, \sigma_0 \approx 0, \alpha_1 \approx 0, \sigma_1 \approx 1$; and the reverse process is defined by denoising from $p_1(\boldsymbol{x}_1) := \mathcal{N}(\boldsymbol{0}, \boldsymbol{I})$ with a parameterized *noise prediction*

---

[2]A prolific dreamer is someone who experiences vivid dreams quite regularly [50], which corresponds to the high-fidelity and diverse results of our method.

*network* $\epsilon_\phi(\boldsymbol{x}_t, t)$ to predict the noise added to a clean data $\boldsymbol{x}_0$, which is trained by minimizing

$$\mathcal{L}_{\mathrm{Diff}}(\phi) := \mathbb{E}_{\boldsymbol{x}_0 \sim q_0(\boldsymbol{x}_0), t \sim \mathcal{U}(0,1), \epsilon \sim \mathcal{N}(\boldsymbol{0}, \boldsymbol{I})}[\omega(t)\|\epsilon_\phi(\alpha_t \boldsymbol{x}_0 + \sigma_t \epsilon) - \epsilon\|_2^2], \quad (1)$$

where $\omega(t)$ is a time-dependent weighting function. After training, we have $p_t \approx q_t$ and thus we can draw samples from $p_0 \approx q_0$. Moreover, the noise prediction network can be used for approximating the *score function* of both $q_t$ and $p_t$ by $\nabla_{\boldsymbol{x}_t} \log q_t(\boldsymbol{x}_t) \approx \nabla_{\boldsymbol{x}_t} \log p_t(\boldsymbol{x}_t) \approx -\epsilon_\phi(\boldsymbol{x}_t, t)/\sigma_t$.

One of the most successful applications of diffusion models is text-to-image generation [41, 36, 38], where the noise prediction model $\epsilon_\phi(\boldsymbol{x}_t, t, y)$ is conditioned on a text prompt $y$. In practice, classifier-free guidance (CFG [15]) is a key technique for trading off the quality and diversity of the samples, which modifies the model by $\hat{\epsilon}_\phi(\boldsymbol{x}_t, t, y) := (1+s)\epsilon_\phi(\boldsymbol{x}_t, t, y) - s\epsilon_\phi(\boldsymbol{x}_t, t, \varnothing)$, where $\varnothing$ is a special "empty" text prompt representing for the unconditional case, and $s > 0$ is the guidance scale. A larger guidance scale usually improves the text-image alignment but reduces diversity.

**Text-to-3D generation by score distillation sampling (SDS) [34].** SDS is an optimization method by distilling pretrained diffusion models, also known as Score Jacobian Chaining (SJC) [55]. It is widely used in text-to-3D generation [34, 55, 20, 29, 55, 4] with great promise. Given a pretrained text-to-image diffusion model $p_t(\boldsymbol{x}_t|y)$ with the noise prediction network $\epsilon_{\mathrm{pretrain}}(\boldsymbol{x}_t, t, y)$, SDS optimizes a single 3D representation with parameter $\theta \in \Theta$, where $\Theta$ is the space of $\theta$ with the Euclidean metric. Given a camera parameter $c$ with a distribution $p(c)$ and a differentiable rendering mapping $\boldsymbol{g}(\cdot, c) : \Theta \to \mathbb{R}^d$, denote $y^c$ as the "*view-dependent prompt*" [34] (i.e., a text prompt with view information), $q_t^\theta(\boldsymbol{x}_t|c)$ as the distribution at time $t$ of the forward diffusion process starting from the rendered image $\boldsymbol{g}(\theta, c)$ with the camera $c$ and 3D parameter $\theta$. SDS optimizes the parameter $\theta$ by solving

$$\min_{\theta \in \Theta} \mathcal{L}_{\mathrm{SDS}}(\theta) := \mathbb{E}_{t,c}\left[(\sigma_t/\alpha_t)\omega(t) D_{\mathrm{KL}}(q_t^\theta(\boldsymbol{x}_t|c) \| p_t(\boldsymbol{x}_t|y^c))\right], \quad (2)$$

where $t \sim \mathcal{U}(0.02, 0.98)$, $\epsilon \sim \mathcal{N}(\boldsymbol{0}, \boldsymbol{I})$, and $\boldsymbol{x}_t = \alpha_t \boldsymbol{g}(\theta, c) + \sigma_t \epsilon$. Its gradient is approximated by

$$\nabla_\theta \mathcal{L}_{\mathrm{SDS}}(\theta) \approx \mathbb{E}_{t,\epsilon,c}\left[\omega(t)(\epsilon_{\mathrm{pretrain}}(\boldsymbol{x}_t, t, y^c) - \epsilon)\frac{\partial \boldsymbol{g}(\theta, c)}{\partial \theta}\right]. \quad (3)$$

Notwithstanding this progress, empirical observations [34] show that SDS often suffers from over-saturation, over-smoothing, and low-diversity issues, which have yet to be thoroughly explained or adequately addressed.

**3D representations.** We employ NeRF [30, 32] (Neural Radiance Fields) and textured mesh [45] as two popular and important types of 3D representations. In particular, NeRF represents 3D objects using a multilayer perceptron (MLP) that takes coordinates in a 3D space as input and outputs the corresponding color and density. Here, $\theta$ corresponds to the parameters of the MLP. Given camera pose $c$, the rendering process $\boldsymbol{g}(\theta, c)$ is defined as casting rays from pixels and computing the weighted sum of the color of the sampling points along each ray to composite the color of each pixel. NeRF is flexible for optimization and is capable of representing extremely complex scenes. Textured mesh [45] represents the geometry of a 3D object with triangle meshes and the texture with color on the mesh surface. Here the 3D parameter $\theta$ consists of the parameters to represent the coordinates of triangle meshes and parameters of the texture. The rendering process $\boldsymbol{g}(\theta, c)$ given camera pose $c$ is defined by casting rays from pixels and computing the intersections between rays and mesh surfaces to obtain the color of each pixel. The textured mesh allows high-resolution and fast rendering with differentiable rasterization.

## 3 Variational Score Distillation

We now present Variational Score Distillation (VSD) (see Sec. 3.1) that learns to sample from a distribution of the 3D scenes. By using 3D parameter particles to represent the target 3D distribution, we derive a principled gradient-based update rule for the particles via the Wasserstein gradient flow (see Sec. 3.2). We further show that SDS is a special case of VSD and constructs an experiment in 2D space to study the optimization algorithm isolated from the 3D representations, explaining the practical issues of SDS both theoretically and empirically (see Sec. 3.3).

### 3.1 Sampling from 3D Distribution as Variational Inference

In principle, given a valid text prompt $y$, there exists a probabilistic distribution of all possible 3D representations. Under a 3D representation (e.g., NeRF) parameterized by $\theta$, such a distribution can

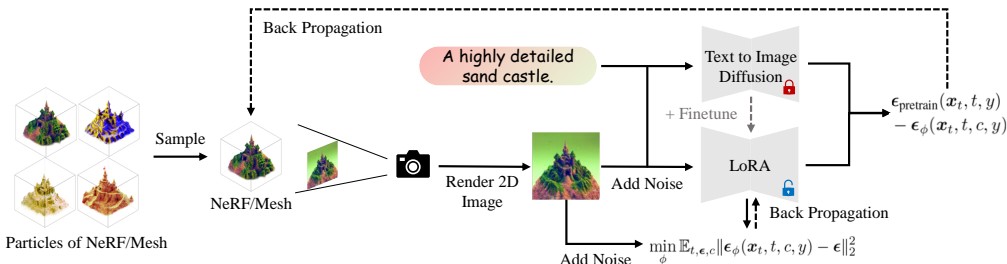

Figure 2: Overview of VSD. The 3D representation is differentiably rendered at a random pose $c$. The rendered image is sent to the pretrained diffusion and the score of the variational distribution (estimated by LoRA) to compute the gradient of VSD. LoRA is also updated on the rendered image.

be modeled as a probabilistic density $\mu(\theta|y)$. Denote $q_0^\mu(\boldsymbol{x}_0|c, y)$ as the (implicit) distribution of the rendered image $\boldsymbol{x}_0 := \boldsymbol{g}(\theta, c)$ given the camera $c$ with the rendering function $\boldsymbol{g}(\cdot, c)$, and denote $p_0(\boldsymbol{x}_0|y^c)$ as the marginal distribution of $t = 0$ defined by the pretrained text-to-image diffusion model with the view-dependent prompt $y^c$. To obtain 3D representations of high visual quality, we propose to optimize the distribution $\mu$ to align the rendered images of its samples with the pretrained diffusion model in all views by solving

$$\min_\mu D_{\mathrm{KL}}(q_0^\mu(\boldsymbol{x}_0|c, y) \,\|\, p_0(\boldsymbol{x}_0|y^c)). \tag{4}$$

This is a typical variational inference problem that uses the variational distribution $q_0^\mu(\boldsymbol{x}_0|c, y)$ to approximate (distill) the target distribution $p_0(\boldsymbol{x}_0|y^c)$.

Directly solving problem (4) is hard because $p_0$ is rather complex and the high-density regions of $p_0$ may be extremely sparse in high dimension [47]. Inspired by the success of diffusion models [14, 49], we construct a series of optimization problems with different diffused distributions indexed by $t$. As $t$ increases to $T$, the optimization problem becomes easier because the diffused distributions get closer to the standard Gaussian. We simultaneously solve an ensemble of these problems (termed as *variational score distillation* or VSD) as follows:

$$\mu^* := \arg\min_\mu \mathbb{E}_{t,c}\left[(\sigma_t/\alpha_t)\omega(t)D_{\mathrm{KL}}(q_t^\mu(\boldsymbol{x}_t|c, y) \,\|\, p_t(\boldsymbol{x}_t|y^c))\right], \tag{5}$$

where $q_t^\mu(\boldsymbol{x}_t|c, y) := \int q_0^\mu(\boldsymbol{x}_0|c, y)p_{t0}(\boldsymbol{x}_t|\boldsymbol{x}_0)\mathrm{d}\boldsymbol{x}_0$ and $p_t(\boldsymbol{x}_t|y^c) := \int p_0(\boldsymbol{x}_0|y^c)p_{t0}(\boldsymbol{x}_t|\boldsymbol{x}_0)\mathrm{d}\boldsymbol{x}_0$ are the corresponding noisy distributions at time $t$ with the Gaussian transition $p_{t0}(\boldsymbol{x}_t|\boldsymbol{x}_0) = \mathcal{N}(\boldsymbol{x}_t|\alpha_t\boldsymbol{x}_0, \sigma_t^2\boldsymbol{I})$, and $\omega(t)$ is a time-dependent weighting function.

Compared with SDS that optimizes for the single point $\theta$, VSD optimizes for the whole distribution $\mu$, from which we sample $\theta$. Notably, we prove that introducing the additional KL-divergence for $t > 0$ in VSD does not affect the global optimum of the original problem (4), as shown below.

**Theorem 1** (Global optimum of VSD, proof in Appendix C.4.). *For each $t > 0$, we have*

$$D_{\mathrm{KL}}(q_t^\mu(\boldsymbol{x}_t|c, y) \,\|\, p_t(\boldsymbol{x}_t|y^c)) = 0 \Leftrightarrow q_0^\mu(\boldsymbol{x}_0|c, y) = p_0(\boldsymbol{x}_0|y^c). \tag{6}$$

### 3.2 Update Rule for Variational Score Distillation

To solve problem (5), a direct way can be to train another parameterized generative model for $\mu$, but it may bring much computation cost and optimization complexity. Inspired by previous particle-based variational inference [23, 3, 9] methods, we maintain $n$ 3D parameters[3] $\{\theta\}_{i=1}^n$ as particles and derive a novel update rule for them. Intuitively, we use $\{\theta\}_{i=1}^n$ to "represent" the current distribution $\mu$, and $\theta^{(i)}$ will be samples from the optimal distribution $\mu^*$ if the optimization converges. Such optimization can be realized by simulating an ODE w.r.t. $\theta$, as shown in the following theorem.

**Theorem 2** (Wasserstein gradient flow of VSD, proof in Appendix C). *Starting from an initial distribution $\mu_0$, denote the Wasserstein gradient flow minimizing problem (5) in the distribution (function) space at each time $\tau \geq 0$ as $\{\mu_\tau\}_{\tau \geq 0}$ with $\mu_\infty = \mu^*$. Then we can sample $\theta_\tau$ from $\mu_\tau$ by*

---

[3]We optimize up to $n = 4$ particles due to the computation resource limit. See details in Appendix D.3.

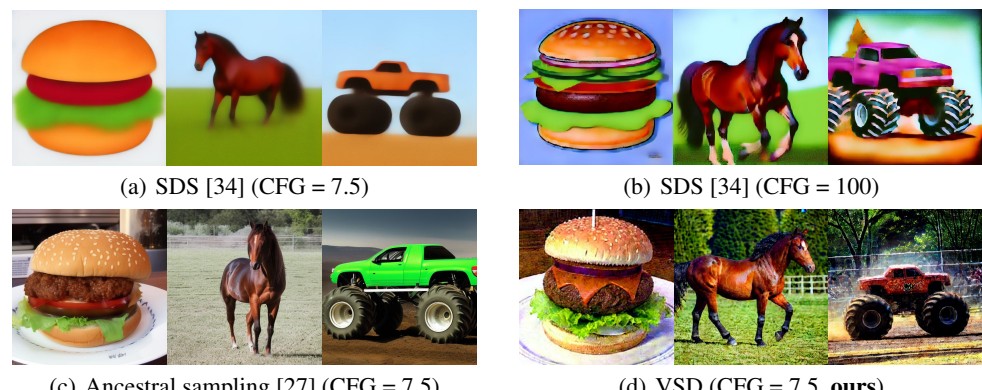

(a) SDS [34] (CFG = 7.5)        (b) SDS [34] (CFG = 100)

(c) Ancestral sampling [27] (CFG = 7.5)      (d) VSD (CFG = 7.5, **ours**)

Figure 3: Samples of different methods in 2D space. Similarly to ancestral sampling, VSD generates realistic images with a common CFG weight of 7.5 and outperforms SDS significantly. The prompts from left to right are *hamburger*, *horse*, and *a monster truck*, respectively. See details in Appendix G.

*firstly sampling $\theta_0 \sim \mu_0(\theta_0|y)$ and then simulating the following ODE:*

$$\frac{\mathrm{d}\theta_\tau}{\mathrm{d}\tau} = -\mathbb{E}_{t,\boldsymbol{\epsilon},c}\left[\omega(t)\Big(\underbrace{-\sigma_t\nabla_{\boldsymbol{x}_t}\log p_t(\boldsymbol{x}_t|y^c)}_{\text{score of noisy real images}} - \underbrace{(-\sigma_t\nabla_{\boldsymbol{x}_t}\log q_t^{\mu_\tau}(\boldsymbol{x}_t|c,y))}_{\text{score of noisy rendered images}}\Big)\frac{\partial\boldsymbol{g}(\theta_\tau,c)}{\partial\theta_\tau}\right], \quad (7)$$

*where $q_t^{\mu_\tau}$ is the corresponding noisy distribution at diffusion time $t$ w.r.t. $\mu_\tau$ at ODE time $\tau$.*

According to Theorem 2, we can simulate the ODE in Eq. (7) for a large enough $\tau$ to approximately sample from the desired distribution $\mu^*$. The ODE involves the score function of noisy real images and that of noisy rendered images at each time[4] $\tau$. The score function of noisy real images $-\sigma_t\nabla_{\boldsymbol{x}_t}\log p_t(\boldsymbol{x}_t|y^c)$ can be approximated by the pretrained diffusion model $\boldsymbol{\epsilon}_{\text{pretrain}}(\boldsymbol{x}_t,t,y^c)$. The score function of noisy rendered images $-\sigma_t\nabla_{\boldsymbol{x}_t}\log q_t^{\mu_\tau}(\boldsymbol{x}_t|c,y)$ is estimated by another noise prediction network $\boldsymbol{\epsilon}_\phi(\boldsymbol{x}_t,t,c,y)$, which is trained on the rendered images by $\{\theta^{(i)}\}_{i=1}^n$ with the standard diffusion objective (see Eq. (1)):

$$\min_\phi \sum_{i=1}^n \mathbb{E}_{t\sim\mathcal{U}(0,1),\boldsymbol{\epsilon}\sim\mathcal{N}(\mathbf{0},\boldsymbol{I}),c\sim p(c)}\left[\|\boldsymbol{\epsilon}_\phi(\alpha_t\boldsymbol{g}(\theta^{(i)},c)+\sigma_t\boldsymbol{\epsilon},t,c,y)-\boldsymbol{\epsilon}\|_2^2\right]. \quad (8)$$

In practice, we parameterize $\boldsymbol{\epsilon}_\phi$ by either a small U-Net [39] or a LoRA (Low-rank adaptation [18, 40]) of the pretrained model $\boldsymbol{\epsilon}_{\text{pretrain}}(\boldsymbol{x}_t,t,y^c)$, and add additional camera parameter $c$ to the condition embeddings in the network. In most cases, we find that using LoRA can greatly improve the fidelity of the obtained samples (e.g., see results in Fig. 1). We believe that it is because LoRA is designed for efficient few-shot fine-tuning and can leverage the prior information in $\boldsymbol{\epsilon}_{\text{pretrain}}$ (the information of both images and text corresponding to $y$).

Note that at each ODE time $\tau$, we need to ensure $\boldsymbol{\epsilon}_\phi$ matches the current distribution $q_t^{\mu_\tau}$. Thus, we optimize $\boldsymbol{\epsilon}_\phi$ and $\theta^{(i)}$ alternately, and each particle $\theta^{(i)}$ is updated by $\theta^{(i)} \leftarrow \theta^{(i)} - \eta\nabla_\theta\mathcal{L}_{\text{VSD}}(\theta^{(i)})$, where $\eta > 0$ is the step size (learning rate). According to Theorem 2, the corresponding gradient is

$$\nabla_\theta\mathcal{L}_{\text{VSD}}(\theta) \triangleq \mathbb{E}_{t,\boldsymbol{\epsilon},c}\left[\omega(t)\left(\boldsymbol{\epsilon}_{\text{pretrain}}(\boldsymbol{x}_t,t,y^c)-\boldsymbol{\epsilon}_\phi(\boldsymbol{x}_t,t,c,y)\right)\frac{\partial\boldsymbol{g}(\theta,c)}{\partial\theta}\right], \quad (9)$$

where $\boldsymbol{x}_t = \alpha_t\boldsymbol{g}(\theta,c)+\sigma_t\boldsymbol{\epsilon}$. We show the approach of VSD in Fig. 3 (see pseudo code in Appendix E).

### 3.3 Comparison with SDS

We now systematically compare VSD with SDS in both theory and practice.

---

[4]Note that we have two variables of time: one is the diffusion time $t \in [0,T]$ and the other is the gradient flow time $\tau$, corresponding to the optimization iteration for each $\theta$.

Table 1: Design space of text-to-3D via 2D diffusion. We highlight the contributions of this paper that improve the fidelity, diversity and ability to generate complex scenes by *, † and ‡ respectively.

| Method | DreamFusion [34] | Magic3D [20] | Fantasia3D [4] | Ours |
|---|---|---|---|---|
| **NeRF Representation** | | | | |
| Resolution* | 64 | 64 | - | 512 |
| Backbone | mipNeRF360 [2] | Instant NGP [32] | - | Instant NGP [32] |
| Initialization‡ | Object | Object | - | Object / Scene initialization |
| **NeRF Training** | | | | |
| Base model | Imagen [41] | eDiff-I [1] | - | Stable Diffusion [38] |
| Number of particles† | 1 | 1 | - | 1∼4 |
| Distillation objective*† | SDS (Eq. (3)) | SDS (Eq. (3)) | - | VSD (Eq. (9)) |
| CFG* | 100 | 100 | - | 7.5 |
| Time schedule* | $\mathcal{U}(0.02, 0.98)$ | $\mathcal{U}(0.02, 0.98)$ | - | $\mathcal{U}(0.02, 0.98) \rightarrow \mathcal{U}(0.02, 0.5)$ |
| **Mesh Representation** | | | | |
| Initialization | - | From NeRF | Handcrafted | From NeRF |
| Texture and geometry | - | Entangled | Disentangled | Disentangled |
| **Mesh Training** | | | | |
| Distillation objective*† | - | SDS (Eq. (3)) | SDS (Eq. (3)) | VSD (Eq. (9)) |
| CFG* | - | 100 | 100 | 7.5 |

**SDS as a special case of VSD.** Theoretically, comparing the update rules of SDS (Eq. (3)) and VSD (Eq. (9)), SDS is a special case of VSD by using a single-point Dirac distribution $\mu(\theta|y) \approx \delta(\theta - \theta^{(1)})$ as the variational distribution (see Appendix C.3 for derivation). In particular, VSD not only employs potentially multiple particles but also learns a parametric score function $\epsilon_\phi$ even for a single particle (i.e., $n = 1$). Empirically, the learned neural network may potentially offer superior generalization ability over the Dirac distribution in SDS, thus it may provide more accurate updating directions in low-density regions. Moreover, by using LoRA, VSD can additionally exploit the text prompt $y$ in the estimation $\epsilon_\phi(x_t, t, c, y)$, while the Gaussian noise $\epsilon$ used in SDS cannot leverage the information from $y$. Thus, VSD may provide samples which are more aligned with the prompt $y$.

**VSD is friendly to CFG.** As VSD aims to *sample* $\theta$ from the optimal $\mu^*$ defined by the pretrained model $\epsilon_{\text{pretrain}}$, the effects by tuning the CFG in $\epsilon_{\text{pretrain}}$ for 3D sampling by VSD are quite similar to 2D sampling by the traditional ancestral sampling methods [14, 27]. Therefore, VSD can tune CFG as flexibly as the classic text-to-image methods, and we use the same setting of CFG (e.g. 7.5) as the common text-to-image generation task for the best performance. To the best of our knowledge, this for the first time addresses the problem in previous SDS [34, 20, 4, 29] that it usually requires a large CFG (i.e., 100).

**VSD vs. SDS in 2D experiments that isolate 3D representations.** To directly compare SDS and VSD, we consider a special case of the rendering function $g(\theta)$ to decouple the optimization algorithm from 3D representations. In particular, we set $g(\theta, c) \equiv \theta$ for any $c$. Then the rendered image $x = g(\theta, c) = \theta$ is the same 2D image as $\theta$. In such a case, optimizing the parameter $\theta$ is equivalent to generating an image in 2D space, thereby independent of the 3D representation. We show the results of different sampling methods in Fig. 3. SDS exhibits failure under both small and large CFG weights. Particularly with the default CFG weight (i.e., 100) used in SDS, the 2D samples share the same issues previously observed in text-to-3D such as over-saturation and over-smoothing [34]. In contrast, VSD demonstrates flexibility in accommodating various CFG weights and produces realistic samples using a normal CFG weight (i.e., 7.5), behaving similarly to ancestral sampling from diffusion models. See more details and analysis in Appendix G.

As other 3D factors are isolated in this comparison, these theoretical and empirical results suggest that the aforementioned practical issues of SDS [34] stem from the oversimplified variational distribution and large CFG employed by SDS. Such results strongly motivate us to employ VSD for text-to-3D generation, where it still substantially and consistently outperforms SDS (see evidence in Sec 5).

## 4  ProlificDreamer

We further present a clear design space for text-to-3D in Sec. 4.1 and systematically study other elements orthogonal to the distillation algorithm in Sec. 4.2. Combining all improvements highlighted in Tab. 1, we arrive at *ProlificDreamer*, an advanced text-to-3D approach.

| DreamFusion | Magic3D | Fantasia3D | Ours |
|---|---|---|---|

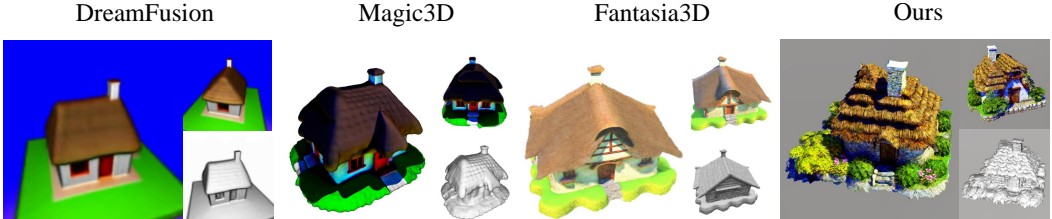

A 3D model of an adorable cottage with a thatched roof.

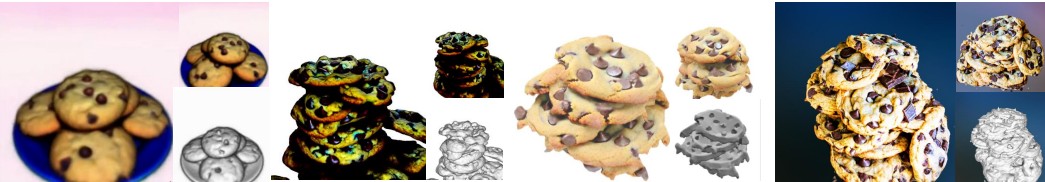

A plate piled high with chocolate chip cookies.

Figure 4: Comparison with baselines. Our results have higher fidelity and more details.

## 4.1 Design Space of Text-to-3D Generation

We adopt the two-stage approach [20], with several improvements in the design space of text-to-3D generation as summarized in Table 1. Specifically, in the first stage, we optimize a high-resolution (e.g., $512$) NeRF by VSD to utilize its high flexibility for generating scenes with complex geometry. In the second stage, we use DMTet [45] to extract textured mesh from the NeRF obtained in the first stage, and further fine-tune the textured mesh for high-resolution details. The second stage is optional because both NeRF and mesh have their own advantages in representing 3D content and are preferred in certain cases. Nevertheless, ProlificDreamer can generate both high-fidelity NeRFs and meshes.

## 4.2 3D Representation and Training

We systematically study other elements orthogonal to the algorithmic formulation. Specifically, we propose a high rendering resolution of $512 \times 512$ during training and an annealed distilling time schedule to improve the visual quality. We also carefully design a scene initialization, which is crucial for complex scene generation.

**High-resolution rendering for NeRF training.** We choose Instant NGP [32] for efficient high-resolution rendering and optimize NeRF with up to 512 training resolution using VSD. By applying VSD, we obtain high-fidelity NeRFs with resolutions varying from $64$ to $512$.

**Scene initialization for NeRF training.** We initialize the density for NeRF as $\sigma_{\text{init}}(\boldsymbol{\mu}) = \lambda_\sigma \left(1 - \frac{||\boldsymbol{\mu}||_2}{r}\right)$, where $\lambda_\sigma$ is the density strength, $r$ is the density radius, and $\boldsymbol{\mu}$ is the coordinate. For object-centric scenes, we follow the *object-centric initialization* used in Magic3D [20] with $\lambda_\sigma = 10$ and $r = 0.5$; For complex scenes, we propose *scene initialization* by setting $\lambda_\sigma = -10$ to make the density "hollow" and $r = 2.5$ that encloses the camera. We show in Appendix D that the scene initialization can help to generate high-fidelity complex scenes without other modifications to the existing algorithm. In addition, we can further add a centric object to the complex scene by using object-centric initialization for $||\boldsymbol{\mu}||_2 < 5/6$ and scene initialization for others, where the hyperparameter $5/6$ ensures the initial density function is continuous.

**Annealed time schedule for score distillation.** We utilize a simple two-stage annealing of time step $t$ in the score distillation objective, suitable for both SDS (Eq. (3)) and VSD (Eq. (9)). For the first several steps we sample time steps $t \sim \mathcal{U}(0.02, 0.98)$ and then anneal into $t \sim \mathcal{U}(0.02, 0.50)$. The key insight is that, essentially, we aim to match the original $q_0^\mu(\boldsymbol{x}_0|c, y)$ with $p_0(\boldsymbol{x}_0|y^c)$. The KL-divergence for larger $t$ can provide reasonable optimization direction during the early stage of training. During training, while $\boldsymbol{x}^*$ is approaching the support of $p_0(\boldsymbol{x}^*|y^c)$, a smaller $t$ can narrow the gap between $p_t(\boldsymbol{x}^*|y^c)$ and $p_0(\boldsymbol{x}^*|y^c)$, and provide elaborate details aligning with $p_0(\boldsymbol{x}^*|y^c)$.

**Mesh representation and fine-tuning.** We adopt a coordinate-based hash grid encoder inherited from NeRF stage to represent the mesh texture. We follow Fantasia3D [4] to disentangle the

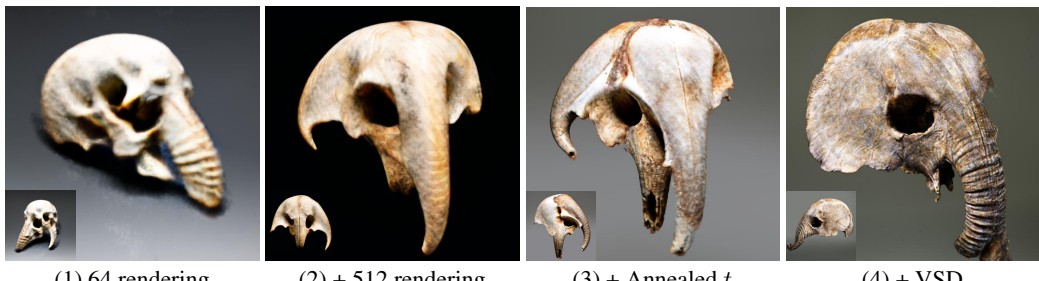

| (1) 64 rendering | (2) + 512 rendering | (3) + Annealed $t$ | (4) + VSD |

Figure 5: Ablation study of proposed improvements for high-fidelity NeRF generation. The prompt is *an elephant skull*. (1) The common setting [34, 20] adopts 64 rendering resolution and SDS loss. (2) We improve the generated quality by increasing the rendering resolution. (3) Annealed time schedule adds more details to the generated result. (4) VSD makes the results even better with richer details.

optimization of geometry and texture by first optimizing the geometry using the normal map and then optimizing the texture. In our initial experiments, we find that optimizing geometry with VSD provides no more details than using SDS. This may be because the mesh resolution is not large enough to represent high-frequency details. Thus, we optimize geometry with SDS for efficiency. But unlike Fantasia3D [4], our texture optimization is supervised by VSD with CFG = 7.5 with the annealed time schedule, which can provide more details than SDS.

## 5 Experiments

### 5.1 Results of ProlificDreamer

We show the generated results of ProlificDreamer in Fig. 1(a) and 1(b), including high-fidelity mesh and NeRF results. All the results are generated by VSD. For all experiments without mentioned, VSD uses $n = 1$ particle for a fair comparison with SDS. In Fig. 1(c), we also demonstrate VSD can generate diverse results, showing that different particles in a round are diverse (with $n = 4$).

**Object-centric generation.** We compare our method with three SOTA baselines, DreamFusion [34], Magic3D [20] and Fantasia3D [4]. All of the baselines are based on SDS. Since none of them is open-sourced, we use the figures from their papers. As shown in Fig. 4, ProlificDreamer generates 3D objects with higher fidelity and more details, which demonstrates the effectiveness of our method.

In appendix, we add user study (Section J) and quantitative results (Section K) of ProlificDreamer against the baselines to demonstrate the effectiveness of our method.

**Large scene generation.** As shown in Fig. 1(b), our method can generate $360°$ scenes with high-fidelity and fine details. The depth map shows that the scenes have geometry instead of being a $360°$ textured sphere, verifying that with our scene initialization alone we can generate high-fidelity large scenes without much modification to existing components. See more results in Appendix B.

### 5.2 Ablation Study

**Ablation on NeRF training.** Fig. 5 provides the ablation on NeRF training. Starting from the common setting [34, 20] with 64 rendering resolution and SDS loss, we ablate our proposed improvements step by step, including increasing resolution, adding annealed time schedule, and adding VSD all improve the generated results. It demonstrates the effectiveness of our proposed components. We provide more ablation on large scene generation in Appendix D, with a similar conclusion.

**Ablation on mesh fine-tuning.** We ablate between SDS and VSD on mesh fine-tuning, as shown in Appendix D. Fine-tuning texture with VSD provides higher fidelity than SDS. As the fine-tuned results of textured mesh are highly dependent on the initial NeRF, getting a high-quality NeRF at the first stage is crucial. Note that the provided results of both VSD and SDS in mesh fine-tuning are based on and benefit from the high-fidelity NeRF results in the first stage by our VSD.

**Ablation on CFG.** We perform ablation to explore how CFG affects generation diversity. We find that smaller CFG encourages more diversity. Our VSD works well with small CFG and provides

considerable diversity, while SDS cannot generate plausible results with small CFG (e.g., 7.5), which limits its ability to generate diverse results. Results and more details are shown in Appendix H.

## 6 Related Works

**Diffusion models.**  Score-based generative model [48, 49] and diffusion models [14] have shown great performance in image synthesis [14, 7]. Recently, large-scale diffusion models have shown great performance in text-to-image synthesis [36, 41, 38], which provides an opportunity to utilize it for zero-shot text-to-3D generation.

**Text-to-3D generation.**  DreamField [19] proposes a text-to-3D method using CLIP [35] guidance. DreamFusion [34] proposes a text-to-3D method using 2D diffusion models. Score Jacobian Chaining (SJC) [55] derives the training objective of text-to-3D using a 2D diffusion model from another theoretical basis. Magic3D [20] extends text-to-3D to a higher resolution with mesh [45] representation. Latent-NeRF [29] optimizes NeRF in latent space. Fantasia3D [4] optimizes a mesh with DMTet [45] from scratch. Although Fantasia3D achieves remarkable zero-shot text-to-3D generation, it requires user-provided shape guidance for the generation of complex geometry. [17] propose score debiasing and prompt debiasing to mitigate multiface problem and is orthogonal to our work. TextMesh [52] is contemporary with us and proposes a different pipeline for high-fidelity text-to-3D generation. 3DFuse [44] proposes to incorporate 3D awareness into 2D diffusion for better 3D consistency in text-to-3D generation. In addition, adjusting time schedule has also been discussed in previous works [12, 55]. However, previous works either require carefully devising the schedule [12] or perform inferior to the simple random time schedule [55]. Instead, our 2-stage annealing schedule is easy to train and achieves better generation quality than the random time schedule.

**Text-driven large scene generation.**  Text2Room [16] generates indoor rooms from a given prompt. However, it uses additional monocular depth estimation models as prior, and we do not use any additional models. Our method generates the wild text prompt and uses only a text-to-image diffusion model. Set-the-scene [5] is a contemporary work with us aimed at large scene generation with a different pipeline. Overall, ProlificDreamer uses an advanced optimization algorithm, i.e, VSD with our proposed two-stage annealed time schedule, which has a significant advantage over the previous SDS / SJC (see Appendix C.3 for details). As a result, ProlificDreamer achieves high-fidelity NeRF with rich structure and complex effects (e.g., smoke and drops) and photo-realistic mesh results.

## 7 Conclusion

In this work, we systematically study the problem of text-to-3D generation. In terms of the algorithmic formulation, we propose variational score distillation (VSD), a principled particle-based variational framework that treats the 3D parameter as a random variable and infers its distribution. VSD naturally generalizes SDS in the variational formulation and addresses the practical issues of SDS observed before. With other orthogonal improvements to 3D representations, our overall approach, ProlificDreamer, can generate high-fidelity NeRF and photo-realistic textured meshes.

**Limitations and broader impact.**  Although ProlificDreamer achieves remarkable text-to-3D results, the generation takes hours of time, which is much slower than image generation by a diffusion model. Although large scene generation can be achieved with our scene initialization, the camera poses during training are regardless of the scene structure, which may be improved by devising an adaptive camera pose range according to the scene structure for better-generated details. Moreover, due to the limited expressiveness of the base 2D model, the generation for complex prompts may fail, and the generated samples may have multi-face Janus problem [57] in some cases because of the poor text-image alignment for the view-dependent prompts. In addition, content creation by generative models may cause harmful social impacts on the labor market. Also, like other generative models, our method may be utilized to generate fake and malicious contents, which needs more caution.

## Acknowledgement

This work was supported by NSF of China Projects (Nos. 62061136001, 61620106010, 62076145, U19B2034, U1811461, U19A2081, 6197222); Beijing Outstanding Young Scientist Program NO.

BJJWZYJH012019100020098; a grant from Tsinghua Institute for Guo Qiang; the High Performance Computing Center, Tsinghua University; and the Research Funds of Renmin University of China (22XNKJ13). J.Z was also supported by the XPlorer Prize. C. Li was also sponsored by Beijing Nova Program (No. 20220484044).

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

# A Additional Experiment Results of 3D Textured Meshes

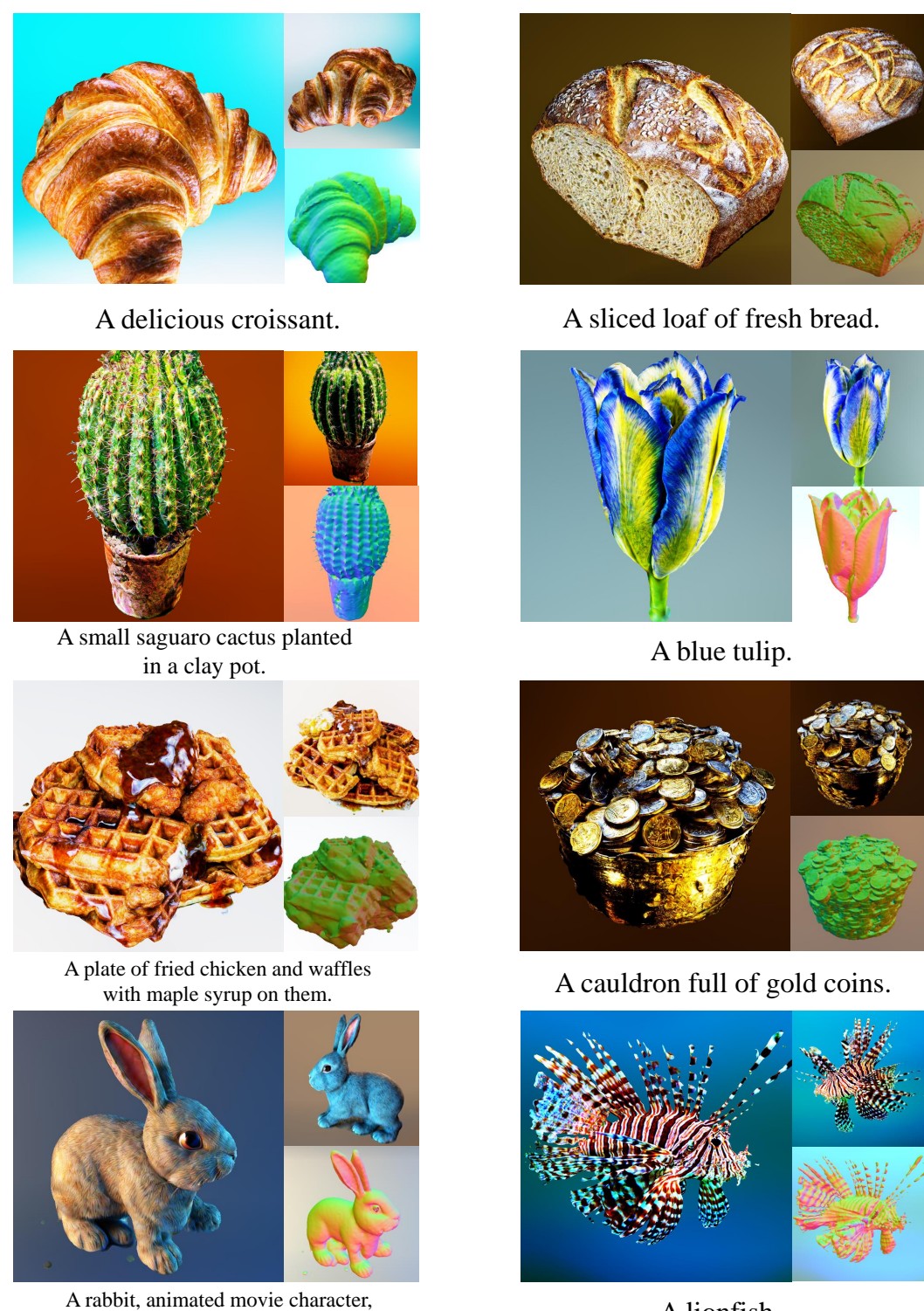

A delicious croissant.

A sliced loaf of fresh bread.

A small saguaro cactus planted
in a clay pot.

A blue tulip.

A plate of fried chicken and waffles
with maple syrup on them.

A cauldron full of gold coins.

A rabbit, animated movie character,
high detail 3d model.

A lionfish.

Figure 6: More results of ProlificDreamer of 3D textured meshes.

We provide more results of ProlificDreamer of 3D textured meshes in Fig. 6 and Fig. 7.

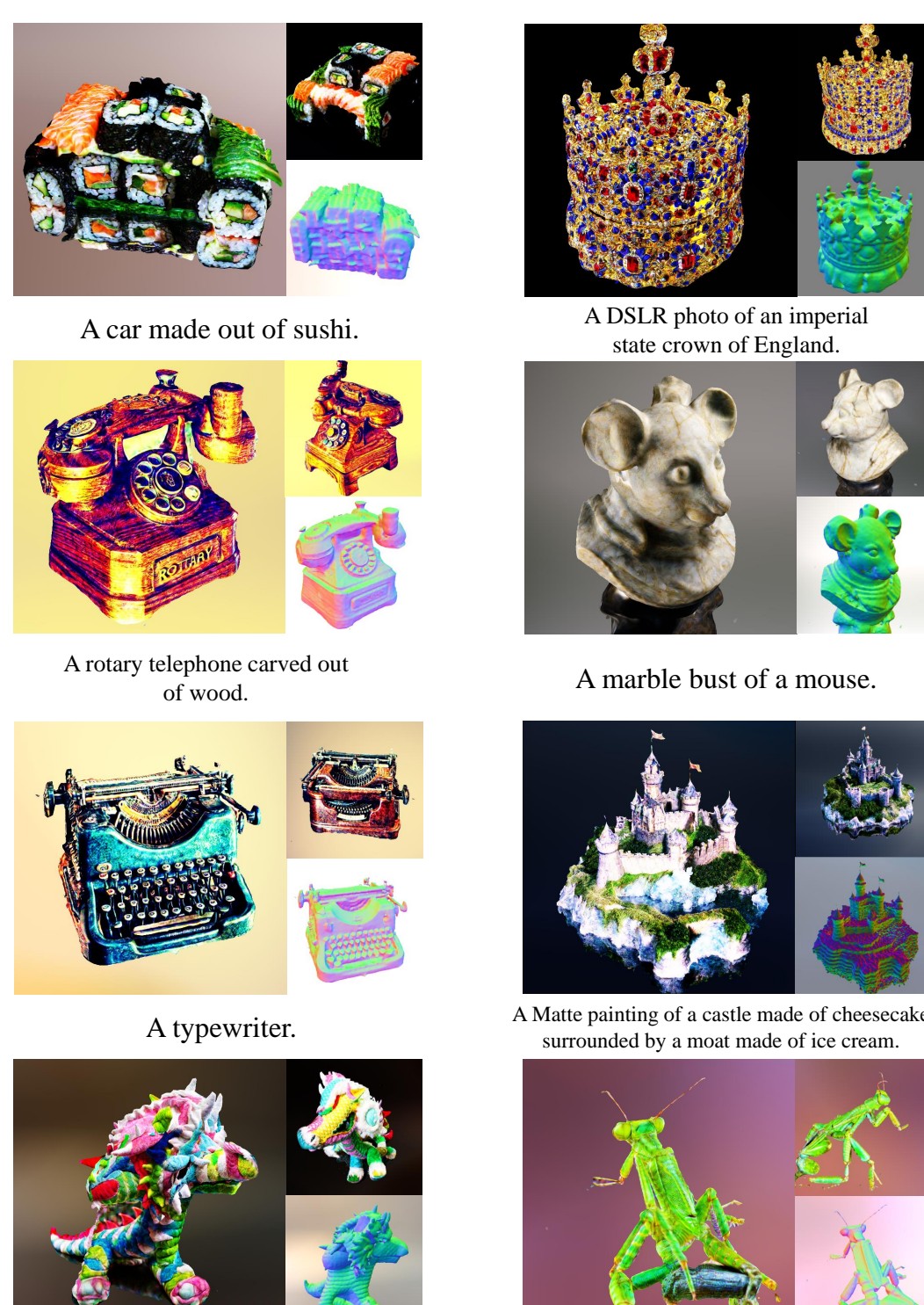

A car made out of sushi.

A DSLR photo of an imperial
state crown of England.

A rotary telephone carved out
of wood.

A marble bust of a mouse.

A typewriter.

A Matte painting of a castle made of cheesecake
surrounded by a moat made of ice cream.

A plush dragon toy.

A praying mantis wearing roller.

Figure 7: More results of ProlificDreamer of 3D textured meshes.

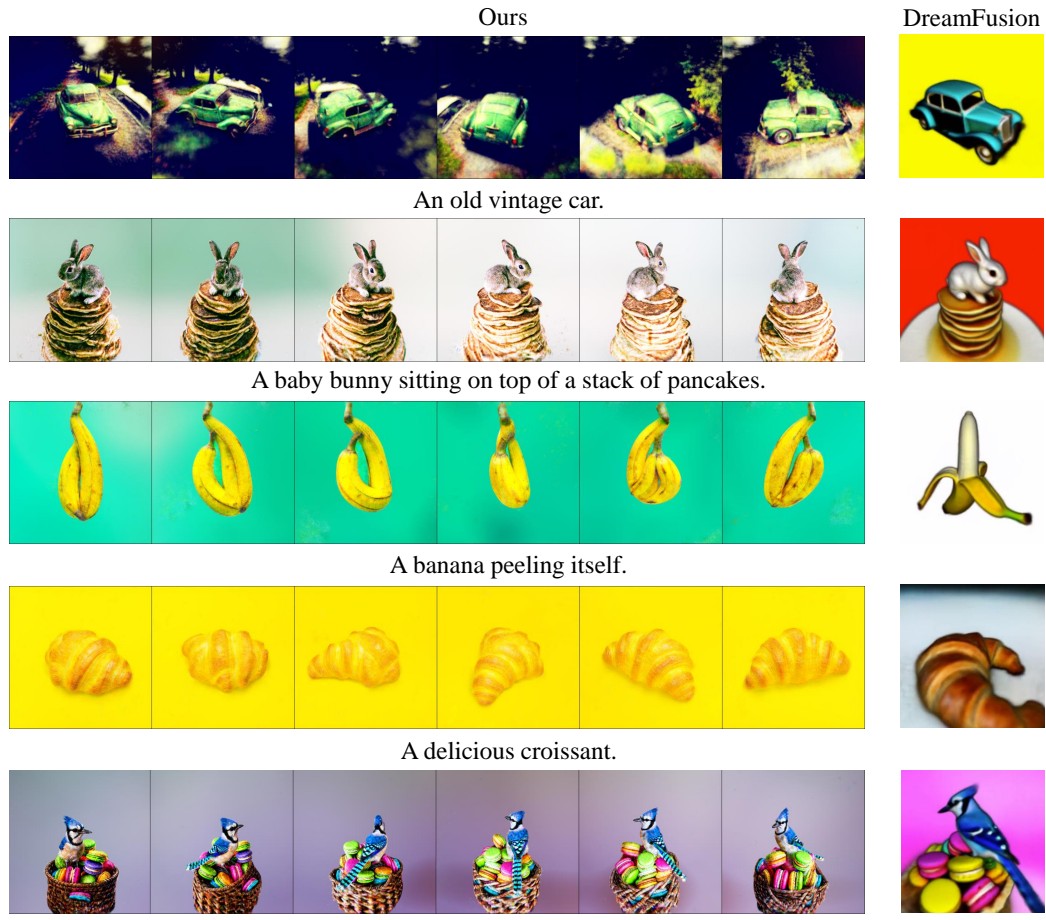

Ours                                                                                           DreamFusion

An old vintage car.

A baby bunny sitting on top of a stack of pancakes.

A banana peeling itself.

A delicious croissant.

A blue jay standing on a large basket of rainbow macarons.

Figure 8: Results of high quality NeRF from ProlificDreamer. Compared with DreamFusion, our ProlificDreamer generates better NeRF results in terms of fidelity and details, which demonstrates the effectiveness of our VSD against SDS.

## B   Additional Experiment Results of 3D NeRF

Here, we provide more generated results in Fig. 8 of high quality NeRF. We compare with Dream-Fusion, as DreamFusion also uses NeRF representation. It can be seen from the figure that Pro-lificDreamer generates better NeRF results in terms of fidelity and details, which demonstrates the effectiveness of our VSD against SDS.

In Fig. 9, we provide more results of large scenes with $360°$ environment using our scene initialization.

In Fig. 10, we provide more results of multiple particle experiments.

## C   Theory of Variational Score Distillation

### C.1   Particle-based Variational Inference

Particle-based variational inference (ParVI) [23, 6, 3, 56, 25, 9] aims to draw (particle) samples from a desired distribution by minimizing the KL-divergence between particle samples and the desired distribution, and is widely-used in Bayesian inference [23, 10, 21]. Specifically, denote $\mathcal{P}(\Theta)$ as the set containing all the distributions on a support space $\Theta$ with Euclidean distance, and $\mathbb{W}_2(\Theta) \coloneqq \{\mu \in \mathcal{P}(\Theta) : \exists \theta_0 \in \Theta \text{ s.t. } \mathbb{E}_{\mu(\theta)}[\|\theta - \theta_0\|^2] < +\infty\}$ as the 2-Wasserstein space [53] equipped with the Wasserstein 2-distance [53]. Given a desired distribution $\mu^* \in \mathcal{P}(\Theta)$, ParVI

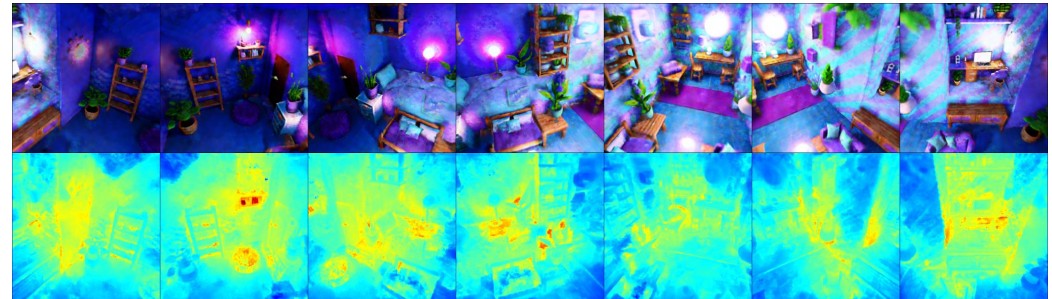

Figure 9: Result of high quality NeRF from ProlificDreamer, a large scene with $360°$ environment using our scene initialization. The prompt here is *Small lavender isometric room, soft lighting, unreal engine render, voxels.*

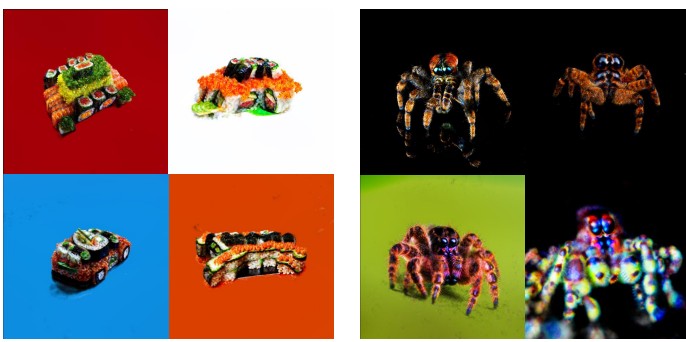

A car made out of sushi.          A tarantula, highly detailed.

Figure 10: More results of multi-particle experiments in NeRF.

starts with some ($n$) particles $\{\theta_0^{(i)}\}_{i=1}^n$ sampled from an initial distribution $\mu_0 \in \mathcal{P}(\Theta)$ and update these particles with a vector field $\boldsymbol{v}_\tau(\cdot)$ on $\Theta$ by $\mathrm{d}\theta_\tau^{(i)} = \boldsymbol{v}_\tau(\theta_\tau^{(i)})\mathrm{d}\tau$ at each time $\tau \geq 0$, such that the distribution $\mu_\tau$ of $\{\theta_\tau^{(i)}\}_{i=0}^n$ converges to $\mu^*$ as $\tau \to \infty$ and $n \to \infty$. Typically, the measure $\mu_\tau$ follows the "steepest descending curves" of a functional on $\mathbb{W}_2(\Theta)$, which is known as the *Wasserstein gradient flow*. A common setting is to define the functional of $\mu_\tau$ as the KL-divergence $D_{\mathrm{KL}}(\mu_\tau \parallel \mu^*)$, and then $\mu_\tau$ follows $\partial_\tau \mu_\tau = -\nabla_\theta \cdot (\mu_\tau \nabla_\theta \log \frac{\mu^*}{\mu_\tau})$, and the corresponding vector field is $\boldsymbol{v}_\tau(\theta_\tau) = \nabla_\theta \log \frac{\mu^*}{\mu_\tau}$. However, directly computing the vector field is intractable because it is non-trivial to compute the time-dependent score function $\nabla_\theta \log \mu_\tau(\theta)$. To tackle this issue, traditional ParVI methods either restrict the functional gradient within RKHS and leverage analytical kernels to approximate the vector field [23, 3, 22], or learn a neural network to estimate the vector field [8]; but all of these methods are hard to scale up to high-dimensional data such as images.

## C.2 VSD as Particle-based Variational Inference

In this section, we formally propose the theoretical guarantee of VSD.

Denote $\mathbb{W}_2(\Theta)$ as the 2-Wasserstein space [53] of the probability distributions defined on the parameter space $\Theta$. For any distribution $\mu \in \mathbb{W}_2(\Theta)$ with a corresponding random variable $\theta \sim \mu(\theta|y)$, denote the implicit distribution for $\boldsymbol{x}_0 := \boldsymbol{g}(\theta, c)$ as $q_0^\mu(\boldsymbol{x}_0|c, y)$, and let $q_t^\mu(\boldsymbol{x}_t|c, y) := \int q_0^\mu(\boldsymbol{x}_0|c, y)q_{t0}(\boldsymbol{x}_t|\boldsymbol{x}_0)\mathrm{d}\boldsymbol{x}_0$ be the marginal distribution of the random variable $\boldsymbol{x}_t$ during the diffusion process. We want to optimize the distribution $\mu(\theta|y)$ in the function space $\mathbb{W}_2(\Theta)$ by distilling a pretrained diffusion model $p_t(\boldsymbol{x}_t|y^c)$ via minimizing the following functional of $\mu$:

$$\min_{\mu \in \mathbb{W}_2(\Theta)} \mathcal{E}[\mu] := \mathbb{E}_{t,\boldsymbol{\epsilon},c}\left[\frac{\sigma_t}{\alpha_t}\omega(t)D_{\mathrm{KL}}(q_t^\mu(\boldsymbol{x}_t|c, y) \parallel p_t(\boldsymbol{x}_t|y^c))\right]. \tag{10}$$

Compared with the vanilla SDS in Eq. (2) which optimizes the parameter $\theta$ in the parameter space $\Theta$, here we aim to optimize the distribution (measure) $\mu(\theta|y)$ in the function space $\mathbb{W}_2(\Theta)$. Inspired by previous ParVI techniques [23, 3, 56], we firstly derive the gradient flow minimizing $\mathcal{E}[\mu]$ in $\mathbb{W}_2(\Theta)$ and the update rule of each particle $\theta \sim \mu(\theta)$, as shown in the following theorem.

**Theorem 3** (Wasserstein gradient flow of score distillation). *Starting from an initial distribution $\mu_0(\theta|y)$, the gradient flow $\{\mu_\tau\}_{\tau \geq 0}$ minimizing $\mathcal{E}[\mu_\tau]$ on $\mathbb{W}_2(\Theta)$ at each time $\tau \geq 0$ satisfies*

$$\frac{\partial \mu_\tau(\theta|y)}{\partial \tau} = -\nabla_\theta \cdot \left[ \mu_\tau(\theta|y) \mathbb{E}_{t,\epsilon,c} \left[ \sigma_t \omega(t) \left( \nabla_{\boldsymbol{x}_t} \log p_t(\boldsymbol{x}_t|y^c) - \nabla_{\boldsymbol{x}_t} \log q_t^{\mu_\tau}(\boldsymbol{x}_t|c,y) \right) \frac{\partial \boldsymbol{g}(\theta,c)}{\partial \theta} \right] \right],$$
(11)

*and the corresponding update rule for each $\theta_\tau \sim \mu_\tau$ satisfies*

$$\frac{\mathrm{d}\theta_\tau}{\mathrm{d}\tau} = -\mathbb{E}_{t,\epsilon,c} \left[ \omega(t) \left( \underbrace{-\sigma_t \nabla_{\boldsymbol{x}_t} \log p_t(\boldsymbol{x}_t|y^c)}_{\approx \boldsymbol{\epsilon}_{pretrain}(\boldsymbol{x}_t,t,y^c)} - \underbrace{(-\sigma_t \nabla_{\boldsymbol{x}_t} \log q_t^{\mu_\tau}(\boldsymbol{x}_t|c,y))}_{\approx \boldsymbol{\epsilon}_\phi(\boldsymbol{x}_t,t,c,y)} \right) \frac{\partial \boldsymbol{g}(\theta_\tau,c)}{\partial \theta_\tau} \right]. \quad (12)$$

Theorem 3 shows that by letting the random variable $\theta_\tau \sim \mu_\tau(\theta_\tau|y)$ move across the ODE trajectory in Eq. (12), its underlying distribution $\mu_\tau$ will move by the direction of the steepest descent that minimizes $\mathcal{E}[\mu]$. Therefore, to obtain samples (in $\Theta$) from $\mu^* = \arg\min_\mu \mathcal{E}[\mu]$, we can simulate the ODE in Eq. (12) by estimating two score functions $\nabla_{\boldsymbol{x}_t} \log p_t(\boldsymbol{x}_t|y^c)$ and $\nabla_{\boldsymbol{x}_t} \log q_t^{\mu_\tau}(\boldsymbol{x}_t|c,y)$ at each ODE time $\tau$, which corresponds to the VSD objective in Eq. (9).

### C.3 Discussions of SDS / SJC and VSD

There are three main differences between our proposed VSD and previous SDS [34] / SJC [55]: **(1)** VSD can optimize multiple ($n \geq 1$) 3D objects, while SDS / SJC only optimizes a single ($n = 1$) 3D object. **(2)** VSD can generate high-fidelity results with the common (e.g., 7.5) CFG (for both $n = 1$ and $n > 1$), while SDS / SJC needs extremely large (e.g. 100) CFG. **(3)** SDS / SJC is a special case of VSD by using a single-point Dirac distribution $\mu(\theta|y) \approx \delta(\theta - \theta^{(1)})$ as the variational distribution.

**SDS / SJC as a special case of VSD.** Here we explain SDS / SJC under our variational inference framework. If we directly approximate $\mu(\theta|y) \approx \delta(\theta - \theta^{(1)})$ by its empirical distribution with the single sample $\theta^{(1)}$, then we have $q_t^\mu(\boldsymbol{x}_t|c,y) \approx \mathcal{N}(\boldsymbol{x}_t|\alpha_t \boldsymbol{g}(\theta^{(1)},c), \sigma_t^2 \boldsymbol{I})$ and thus $-\sigma_t \nabla_{\boldsymbol{x}_t} \log q_t^\mu(\boldsymbol{x}_t|c,y) \approx (\boldsymbol{x}_t - \alpha_t \boldsymbol{g}(\theta^{(1)},c))/\sigma_t = \boldsymbol{\epsilon}$, which recovers the vanilla SDS / SJC. Therefore, SDS / SJC is a special case of VSD for the underlying distribution $\mu(\theta|y)$ by the empirical distribution with a single point. Such an approximation has no generalization ability for $\theta \neq \theta^{(1)}$, and thus the updating direction by the score function $-\sigma_t \nabla_{\boldsymbol{x}_t} \log q_t^\mu(\boldsymbol{x}_t|c,y)$ (or equivalently, the Gaussian noise $\boldsymbol{\epsilon}$) may be rather bad at low-density regions, resulting poor samples for the final $\theta^{(1)}$. Instead, VSD regards $\theta^{(1)}$ as samples from the underlying distribution $\mu(\theta|y)$ and trains a neural network $\boldsymbol{\epsilon}_\phi$ to approximate the corresponding score functions, thus can leverage the generalization ability of neural networks for better approximating the underlying distribution $\mu(\theta|y)$. Moreover, by using LoRA, VSD can additionally exploit the text prompt $y$ in the estimation $\boldsymbol{\epsilon}_\phi(\boldsymbol{x}_t,t,c,y)$, while the Gaussian noise $\boldsymbol{\epsilon}$ used in SDS cannot leverage the information from $y$.

**SDS / SJC as mode-seeking, VSD as sampling.** VSD aims to sample $\theta$ from the optimal $\mu^*$, while SDS / SJC aims to find the optimal $\theta^*$ minimizing the objective in Eq. (2). As the global optimum $\theta^*$ in SDS / SJC is the mode of Eq. (2), SDS / SJC is also known as performing mode-seeking [34]. To demonstrate the differences between sampling and mode-seeking, we consider a special case of the rendering function $\boldsymbol{g}(\theta)$ to decouple the optimization algorithm and the 3D representation. In particular, we set $\boldsymbol{g}(\theta,c) \equiv \theta$ for any $c$ and $\theta \in \mathbb{R}^d$, then the rendered image $\boldsymbol{x} = \boldsymbol{g}(\theta,c) = \theta$ is the same 2D image as $\theta$. In such a case, we have $q_0^\mu = \mu$, and it is easy to prove that $\mu^* = p_0$ (according to Theorem 1). For VSD, sampling $\theta$ from $\mu^*$ is corresponding to the traditional ancestral sampling [27] from $p_0$; while for SDS / SJC, we have $\nabla_\theta \mathcal{L}_{\text{SDS}}(\theta) = \mathbb{E}_{t,\epsilon}[\omega(t)\boldsymbol{\epsilon}_{\text{pretrain}}(\boldsymbol{x}_t,t,y^c)] \approx \mathbb{E}_{t,\epsilon}[-\sigma_t \omega(t) \nabla_{\boldsymbol{x}_t} \log p_t(\boldsymbol{x}_t|y^c)]$, and thus the optimal $\theta^*$ is the mode of the "averaged likelihood" of $p_t$ for all $t$. However, it is common that the mode of deep generative models may have poor sample quality [33]. We show in Fig. 3 that under the same CFG (7.5), both VSD and ancestral sampling can generate good samples but the sample quality of SDS is quite poor.

**SDS / SJC requires large CFG, while VSD is friendly to CFG.** As VSD aims to sample $\theta$ from the optimal $\mu^*$ defined by the pretrained model $\boldsymbol{\epsilon}_{\text{pretrain}}$, the effects by tuning the CFG in $\boldsymbol{\epsilon}_{\text{pretrain}}$ for

3D samples $\theta$ by VSD are quite similar to the effects for the 2D samples by the traditional ancestral sampling [14, 27]. Therefore, VSD can tune CFG as flexibly as the classic text-to-image methods, and we use the same setting of CFG (e.g. 7.5) as the common text-to-image generation task for the best performance. To the best of our knowledge, this for the first time addresses the problem in previous SDS [34, 20, 4, 29] that it usually requires a large CFG (i.e., 100). Specifically, SDS (Eq. (3)) uses $(\epsilon_{\text{pretrain}}(\boldsymbol{x}_t, t, y^c) - \epsilon)$ while VSD (Eq. (9))) uses $(\epsilon_{\text{pretrain}}(\boldsymbol{x}_t, t, y^c) - \epsilon_\phi(\boldsymbol{x}_t, t, c, y))$. For example, for the 2D special case of $\boldsymbol{g}(\theta, c) \equiv \theta$, we have $\nabla_\theta \mathcal{L}_{\text{SDS}}(\theta) = \mathbb{E}_{t,\epsilon}[\omega(t)\epsilon_{\text{pretrain}}(\boldsymbol{x}_t, t, y^c)]$ and $\nabla_\theta \mathcal{L}_{\text{VSD}}(\theta) = \mathbb{E}_{t,\epsilon}[\omega(t)(\epsilon_{\text{pretrain}}(\boldsymbol{x}_t, t, y^c) - \epsilon_\phi(\boldsymbol{x}_t, t, c, y))]$. Intuitively, to obtain highly detailed samples, the updating direction for $\theta$ needs to be "fine" and "sharp". As SDS only depends on $\epsilon_{\text{pretrain}}$, it needs a large CFG (= 100) to make sure $\epsilon_{\text{pretrain}}$ to be "sharp" enough; however, large CFG, in turn, reduces the diversity of the results and also hurts the quality. Instead, VSD leverages an additional score function $\epsilon_\phi(\boldsymbol{x}_t, t, c, y)$ to give a more elaborate direction than the zero-mean Gaussian noise, and the updating direction can be rather "fine" and "sharp" due to the difference between $\epsilon_{\text{pretrain}}$ and $\epsilon_\phi$. We empirically find that VSD can obtain much better quality than SDS in both 2D and 3D generation.

## C.4 Proof of Main Theorem

### C.4.1 Proof of Theorem 1

*Proof of Theorem 1.* Denote $\boldsymbol{x}_0^q$ as the random variable following $q_0^\mu(\boldsymbol{x}_0|c, y)$ and $\boldsymbol{x}_t^q$ as the random variable following $q_t^\mu(\boldsymbol{x}_t|c, y)$. By the definition of $q_t^\mu$, we have

$$\boldsymbol{x}_t^q = \alpha_t \boldsymbol{x}_0^q + \sigma_t \epsilon, \tag{13}$$

where $\epsilon \sim \mathcal{N}(\boldsymbol{0}, \boldsymbol{I})$. Therefore, the characteristic functions of $q_t^\mu$ and $q_0^\mu$ satisfy

$$\varphi_{q_t^\mu}(s) = \varphi_{q_0^\mu}(\alpha_t s) \cdot \varphi_{\mathcal{N}(\boldsymbol{0},\boldsymbol{I})}(\sigma_t s) = e^{-\frac{\sigma_t^2 s^2}{2}} \varphi_{q_0^\mu}(\alpha_t s). \tag{14}$$

Similarly, the characteristic functions of $p_t$ and $p_0$ satisfy

$$\varphi_{p_t}(s) = e^{-\frac{\sigma_t^2 s^2}{2}} \varphi_{p_0}(\alpha_t s). \tag{15}$$

Therefore, we have

$$D_{\text{KL}}(q_t^\mu(\boldsymbol{x}_t|c, y) \parallel p_t(\boldsymbol{x}_t|y^c)) = 0 \Leftrightarrow q_t^\mu = p_t \Leftrightarrow \varphi_{q_t^\mu} = \varphi_{p_t} \Leftrightarrow \varphi_{q_0^\mu} = \varphi_{p_0} \Leftrightarrow q_0^\mu = p_0 \tag{16}$$

$\square$

### C.4.2 Proof of Theorem 3

As the gradient flow minimizing $\mathcal{E}[\mu]$ on $\mathbb{W}_2(\Theta)$ satisfies

$$\frac{\partial \mu_\tau}{\partial \tau} = -\nabla_{\mathbb{W}_2} \mathcal{E}[\mu] = \nabla_\theta \cdot \left( \mu_\tau \nabla_\theta \frac{\delta \mathcal{E}[\mu_\tau]}{\delta \mu_\tau} \right), \tag{17}$$

thus we only need to compute the first variation $\frac{\delta \mathcal{E}[\mu]}{\delta \mu}$. We propose the following lemmas for computing $\frac{\delta \mathcal{E}[\mu]}{\delta \mu}$.

**Lemma 1.** *For $p, q \in \mathbb{W}_2(\mathbb{R}^d)$, for any $\boldsymbol{x} \in \mathbb{R}^d$,*

$$\left( \frac{\delta D_{\text{KL}}(q \parallel p)}{\delta q} \right) [\boldsymbol{x}] = \log q(\boldsymbol{x}) - \log p(\boldsymbol{x}) + 1 \tag{18}$$

*Proof.* This is a classic conclusion in particle-based variational inference (e.g., see Sec.3.2. in [3]). $\square$

**Lemma 2.** *For a fixed $c$ and $\boldsymbol{x}_t$,*

$$\frac{\delta q_t^\mu(\boldsymbol{x}_t|c, y)}{\delta \mu}[\theta] = q_{t0}(\boldsymbol{x}_t|\boldsymbol{x}_0) = \mathcal{N}(\boldsymbol{x}_t|\alpha_t \boldsymbol{x}_0, \sigma_t^2 \boldsymbol{I}), \tag{19}$$

*where $\boldsymbol{x}_0 = \boldsymbol{g}(\theta, c)$.*

*Proof of Lemma 2.* By the definition of $q_t^\mu$, we have

$$q_t^\mu(\boldsymbol{x}_t|c,y) = \mathbb{E}_{q_0^\mu(\boldsymbol{x}_0|c,y)}[q_{t0}(\boldsymbol{x}_t|\boldsymbol{x}_0)] = \mathbb{E}_{\mu(\theta|y)}[q_{t0}(\boldsymbol{x}_t|\boldsymbol{g}(\theta,c))], \tag{20}$$

so

$$\frac{\delta q_t^\mu(\boldsymbol{x}_t|c,y)}{\delta\mu}[\theta] = q_{t0}(\boldsymbol{x}_t|\boldsymbol{g}(\theta,c)) = \mathcal{N}(\boldsymbol{x}_t|\alpha_t\boldsymbol{g}(\theta,c),\sigma_t^2\boldsymbol{I}) \tag{21}$$

$\square$

**Lemma 3.**

$$\frac{\delta D_{\mathrm{KL}}(q_t^\mu(\boldsymbol{x}_t|c,y)\parallel p_t(\boldsymbol{x}_t|y^c))}{\delta\mu}[\theta] = \mathbb{E}_{\boldsymbol{\epsilon}}[\log q_t^\mu(\boldsymbol{x}_t|c,y) - \log p_t(\boldsymbol{x}_t|y^c) + 1], \tag{22}$$

*where $\boldsymbol{x}_t = \alpha_t\boldsymbol{g}(\theta,c) + \sigma_t\boldsymbol{\epsilon}$, $\boldsymbol{\epsilon} \sim \mathcal{N}(\boldsymbol{\epsilon}|\boldsymbol{0},\boldsymbol{I})$.*

*Proof of Lemma 3.* By the chain rule of functional derivative, according to Lemma 1 and Lemma 2, we have

$$\frac{\delta D_{\mathrm{KL}}(q_t^\mu(\boldsymbol{x}_t|c,y)\parallel p_t(\boldsymbol{x}_t|y^c))}{\delta\mu}[\theta] = \int \frac{\delta D_{\mathrm{KL}}(q_t^\mu\parallel p_t)}{\delta q_t^\mu}[\boldsymbol{x}_t] \cdot \frac{\delta q_t^\mu(\boldsymbol{x}_t|c,y)}{\delta\mu}[\theta]\mathrm{d}\boldsymbol{x}_t \tag{23}$$

$$= \int (\log q_t^\mu(\boldsymbol{x}_t|c,y) - \log p_t(\boldsymbol{x}_t|y^c) + 1)q_{t0}(\boldsymbol{x}_t|\boldsymbol{x}_0)\mathrm{d}\boldsymbol{x}_t \tag{24}$$

$$= \mathbb{E}_{q_{t0}(\boldsymbol{x}_t|\boldsymbol{x}_0)}\left[\log q_t^\mu(\boldsymbol{x}_t|c,y) - \log p_t(\boldsymbol{x}_t|y^c) + 1\right] \tag{25}$$

$$= \mathbb{E}_{\boldsymbol{\epsilon}}\left[\log q_t^\mu(\boldsymbol{x}_t|c,y) - \log p_t(\boldsymbol{x}_t|y^c) + 1\right], \tag{26}$$

where $\boldsymbol{\epsilon} \sim \mathcal{N}(\boldsymbol{\epsilon}|\boldsymbol{0},\boldsymbol{I})$, $\boldsymbol{x}_0 = \boldsymbol{g}(\theta,c)$, $\boldsymbol{x}_t = \alpha_t\boldsymbol{x}_0 + \sigma_t\boldsymbol{\epsilon}$. $\square$

Below we provide the proof of Theorem 3.

*Proof of Theorem 3.* According to Lemma 3, we have

$$\frac{\delta\mathcal{E}[\mu]}{\delta\mu}[\theta] = \mathbb{E}_{t,\boldsymbol{\epsilon},c}\left[\frac{\sigma_t}{\alpha_t}\omega(t)\left(\log q_t^\mu(\boldsymbol{x}_t|c,y) - \log p_t(\boldsymbol{x}_t|y^c) + 1\right)\right], \tag{27}$$

where $\boldsymbol{x}_t = \alpha_t\boldsymbol{g}(\theta,c) + \sigma_t\boldsymbol{\epsilon}$. So

$$\nabla_\theta\frac{\delta\mathcal{E}[\mu]}{\delta\mu}[\theta] = \mathbb{E}_{t,\boldsymbol{\epsilon},c}\left[\frac{\sigma_t}{\alpha_t}\omega(t)\left(\nabla_{\boldsymbol{x}_t}\log q_t^\mu(\boldsymbol{x}_t|c,y) - \nabla_{\boldsymbol{x}_t}\log p_t(\boldsymbol{x}_t|y^c)\right)\frac{\partial\boldsymbol{x}_t}{\partial\theta}\right] \tag{28}$$

$$= \mathbb{E}_{t,\boldsymbol{\epsilon},c}\left[\sigma_t\omega(t)\left(\nabla_{\boldsymbol{x}_t}\log q_t^\mu(\boldsymbol{x}_t|c,y) - \nabla_{\boldsymbol{x}_t}\log p_t(\boldsymbol{x}_t|y^c)\right)\frac{\partial\boldsymbol{g}(\theta,c)}{\partial\theta}\right]. \tag{29}$$

Thus, the measure $\mu_\tau$ at step $\tau$ during the Wasserstein gradient flow minimizing $\mathcal{E}[\mu]$ on $\mathbb{W}_2(\Theta)$ satisfies

$$\frac{\partial\mu_\tau}{\partial\tau} = -\nabla_{\mathbb{W}_2}\mathcal{E}[\mu] \tag{30}$$

$$= \nabla_\theta \cdot \left(\mu_\tau\nabla_\theta\frac{\delta\mathcal{E}[\mu_\tau]}{\delta\mu_\tau}\right) \tag{31}$$

$$= \nabla_\theta \cdot \left[\mu_\tau(\theta|y)\mathbb{E}_{t,\boldsymbol{\epsilon},c}\left[\sigma_t\omega(t)\left(\nabla_{\boldsymbol{x}_t}\log q_t^{\mu_\tau}(\boldsymbol{x}_t|c,y) - \nabla_{\boldsymbol{x}_t}\log p_t(\boldsymbol{x}_t|y^c)\right)\frac{\partial\boldsymbol{g}(\theta,c)}{\partial\theta}\right]\right]. \tag{32}$$

By the definition of Fokker-Planck equation, the corresponding process of each particle $\theta_\tau$ at time $\tau$ satisfies

$$\frac{\mathrm{d}\theta_\tau}{\mathrm{d}\tau} = \mathbb{E}_{t,\boldsymbol{\epsilon},c}\left[\sigma_t\omega(t)\left(\nabla_{\boldsymbol{x}_t}\log p_t(\boldsymbol{x}_t|y^c) - \nabla_{\boldsymbol{x}_t}\log q_t^{\mu_\tau}(\boldsymbol{x}_t|c,y)\right)\frac{\partial\boldsymbol{g}(\theta,c)}{\partial\theta}\right]. \tag{33}$$

$\square$

# D   Additional Ablation Study

## D.1   Ablation Study on Large Scene Generation

Here we perform an ablation study on large scene generation to validate the effectiveness of our proposed improvements. We start from 64 rendering resolution, with SDS loss and our scene initialization. The results are shown in Figure 11. It can be seen from the figure that, with our scene initialization, the results are with $360°$ surroundings instead of being object-centric. Increasing rendering resolution is slightly beneficial. Adding annealed time schedule improves the visual quality of the results. Replacing SDS with VSD makes the results more realistic with more details.

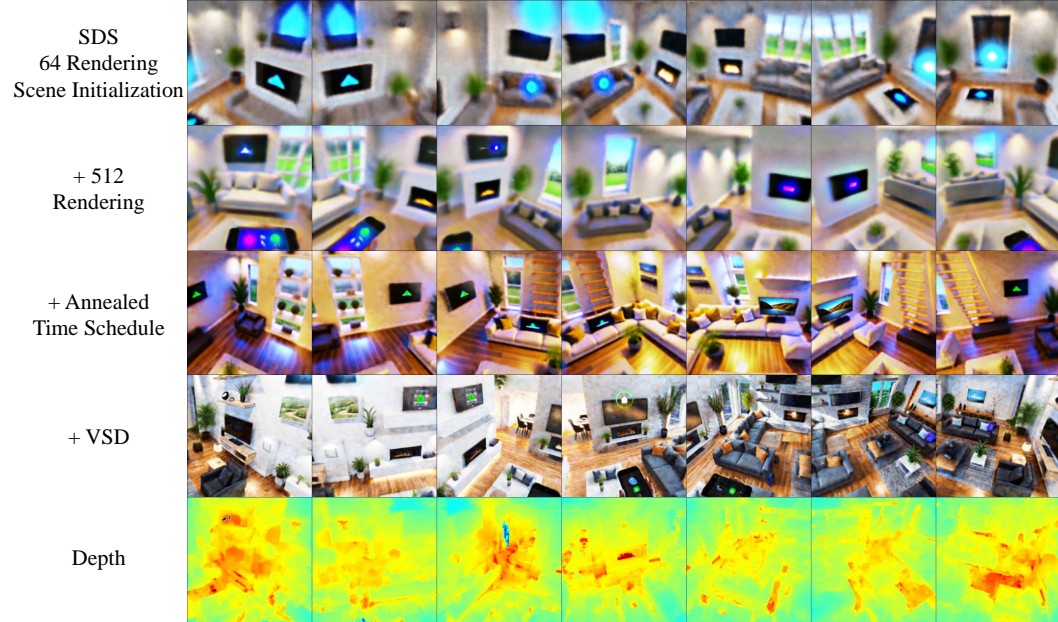

Figure 11: Ablation study on scene generation. With our scene initialization, the results are with $360°$ surroundings instead of being object-centric. Our annealed time schedule and VSD are both beneficial for the generation quality. The prompt here is *Inside of a smart home, realistic detailed photo, 4k.*

## D.2   Ablation Study on Mesh Fine-tuning

Here we provide an ablation study on mesh fine-tuning. Fine-tuning with textured mesh further improves the quality compared to the NeRF result. Fine-tuning texture with VSD provides higher fidelity than SDS. Note that both VSD and SDS in mesh fine-tuning are based on and benefit from the high-fidelity NeRF results by our VSD. And it's crucial to get a high-quality NeRF with VSD at the first stage.

## D.3   Ablation on Number of Particles

Here we provide ablation study on number of particles. We vary the number of particles in $1, 2, 4, 8$ and examine how the number of particles affects the generated results. The CFG of VSD is set as 7.5. The results are shown in Fig. 13. As is shown in the figure, the diversity of the generated results is slightly larger as the number of particles increases. Meanwhile, the quality of generated results is not affected much by the number of particles. Owing to the high computation overhead to optimize 3D representations and limitations on computation resources, we now only test at most 8 particles. We provide a 2D experiment with 2048 particles in Appendix G to demonstrate the scalability of VSD. We leave the experiments of more particles in 3D as future work.

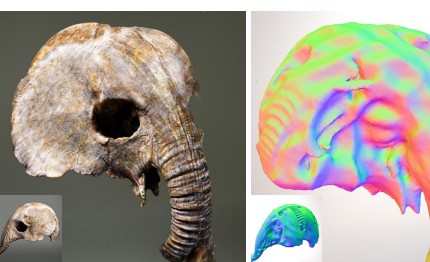 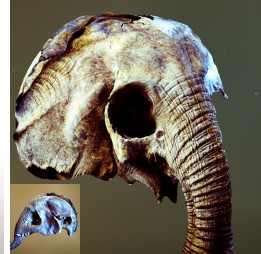 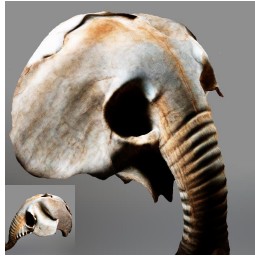

(1) High fidelity NeRF generated by VSD (**ours**).    (2) Extract and fine-tune geometry from NeRF.    (3a) Texture fine-tuned by VSD (**ours**).    (3b) Texture fine-tuned by SDS.

Figure 12: Pipeline of ProlificDreamer along with ablation study of VSD. After generating a high-quality NeRF, we extract and finetune a textured mesh. VSD provides high-fidelity texture, while SDS tends to generate smoother results. Note that both VSD and SDS in mesh fine-tuning are based on and benefit from the high-fidelity NeRF results by our VSD. The prompt here is *an elephant skull*.

---

**Algorithm 1** Variational Score Distillation

---

**Input:** Number of particles $n (\geq 1)$. Large text-to-image diffusion model $\epsilon_{\text{pretrain}}$. Learning rate $\eta_1$ and $\eta_2$ for 3D structures and diffusion model parameters, respectively. A prompt $y$.

1: **initialize** $n$ 3D structures $\{\theta^{(i)}\}_{i=1}^n$, a noise prediction model $\epsilon_\phi$ parameterized by $\phi$.
2: **while** not converged **do**
3:      Randomly sample $\theta \sim \{\theta^{(i)}\}_{i=1}^n$ and a camera pose $c$.
4:      Render the 3D structure $\theta$ at pose $c$ to get a 2D image $\boldsymbol{x}_0 = \boldsymbol{g}(\theta, c)$.
5:      $\theta \leftarrow \theta - \eta_1 \mathbb{E}_{t,\boldsymbol{\epsilon},c} \left[ \omega(t) \left( \boldsymbol{\epsilon}_{\text{pretrain}}(\boldsymbol{x}_t, t, y^c) - \boldsymbol{\epsilon}_\phi(\boldsymbol{x}_t, t, c, y) \right) \frac{\partial \boldsymbol{g}(\theta, c)}{\partial \theta} \right]$
6:      $\phi \leftarrow \phi - \eta_2 \nabla_\phi \mathbb{E}_{t,\epsilon} || \boldsymbol{\epsilon}_\phi(\boldsymbol{x}_t, t, c, y) - \boldsymbol{\epsilon} ||_2^2$.
7: **end while**
8: **return**

---

### D.4 Ablation on Rendering Resolution

Here we provide an ablation study on the rendering resolution during NeRF training with VSD. As shown in Fig. 14, training with a higher resolution produces better results with finer details. In addition, our VSD still provides competitive results under a lower training resolution (128 or 256), which is more computationally efficient than the 512 resolution.

## E    Algorithm for Variational Score Distillation

We provide a summarized algorithm of variational score distillation in Algorithm 1.

We initialize one or several 3D structures $\{\theta^{(i)}\}_{i=1}^n$, a noise prediction model $\epsilon_\phi$ parameterized by $\phi$. At each iteration, we sample a camera pose $c$ from a pre-defined distribution as previous works [34, 20]. Then we render 2D image from 3D structures at pose $c$ with differentiable rendering $\boldsymbol{x}_0 = \boldsymbol{g}(\theta, c)$. To optimize 3D parameters $\theta$, we compute the gradient direction of 2D image and then back propagate to the parameter of NeRF, using VSD as Eq. (9). To model the score of the variational distribution, we train an additional diffusion model $\boldsymbol{\epsilon}_\phi$ parameterized by LoRA. We optimize Eq. (8) to train LoRA after optimization of 3D parameters, using the rendered image $\boldsymbol{x}_0 = \boldsymbol{g}(\theta, c)$ and pose $c$. Note that, at each iteration, we perform differentiable rendering only once but use the rendered image twice for both computing gradient direction with VSD and training LoRA. Thus the computation cost will not increase much compared to SDS.

## F    More Details on Implementation and Hyper-Parameters

**Training details.** We use v-prediction [42] to train our additional diffusion model $\boldsymbol{\epsilon}_\phi$. The camera pose $c$ is fed into a 2-layer MLP and then added to timestep embedding at each U-Net block. NeRF/mesh and LoRA batch sizes are set to 1 owing to the computation limit. A larger batch size of

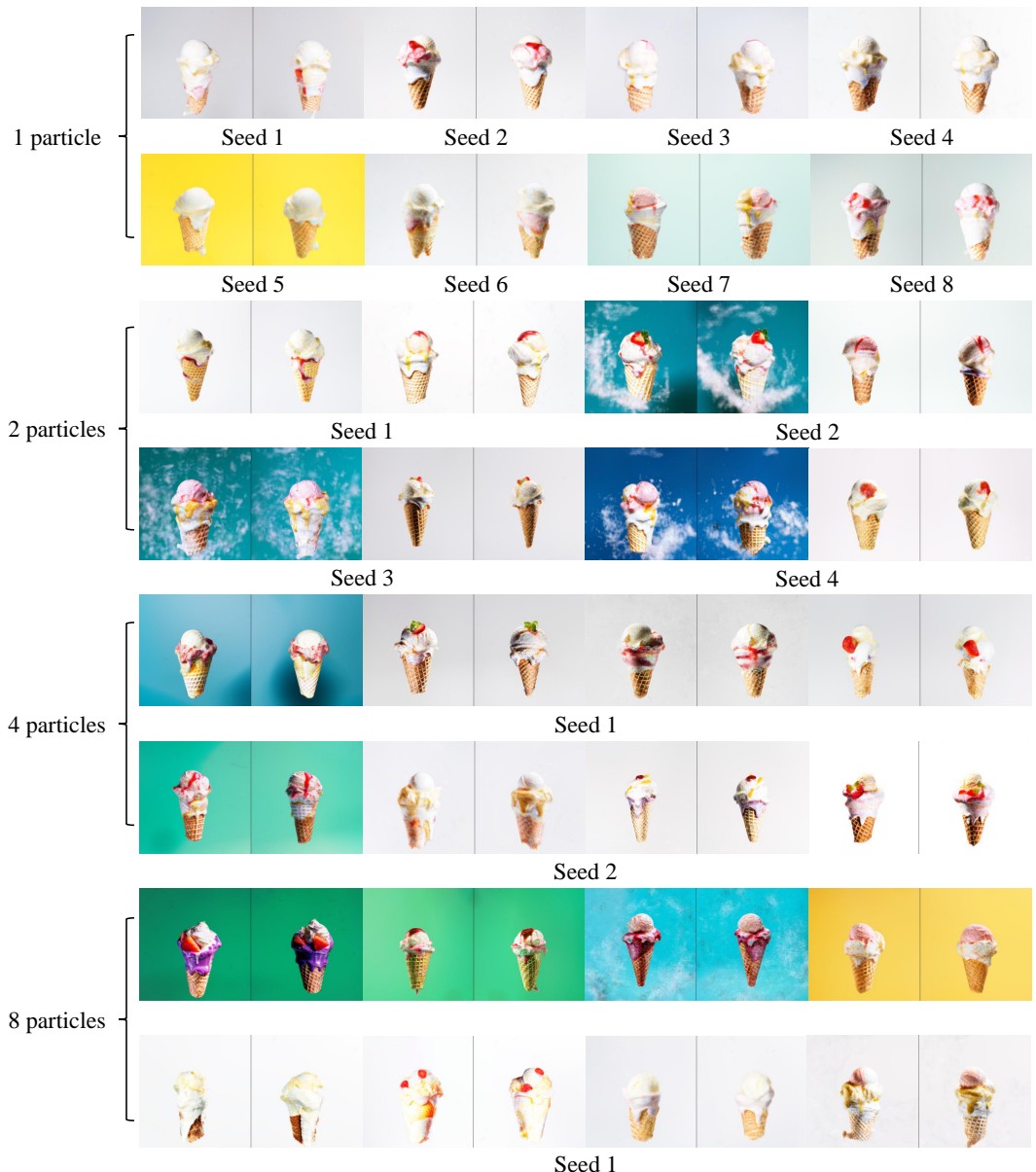

Figure 13: Ablation on how number of particles affects the results. The diversity of the generated results is slightly larger as the number of particles increases. The quality of generated results is not affected much by the number of particles. The prompt is *A high quality photo of an ice cream sundae*.

NeRF/mesh may improve the generated quality and we leave it in future work. The learning rate of LoRA is 0.0001 and the learning rate of hash grid encoder is 0.01. We render in RGB color space for high-resolution synthesis, unlike [29, 55] that render in latent space. In addition, we set $\omega(t) = \sigma_t^2$. For most experiments, we only use $n = 1$ particle for VSD to reduce the computation time (and we only use a batch size of 1, due to the computation resource limits). For NeRF rendering, we sample 96 points along each ray, with 64 samples in coarse stage and 32 samples in fine stage. We choose a single-layer MLP to decode the color and volumetric density from the hash grid encoder as previous work [20].

For object-centric scenes, we set the camera radius as $\mathcal{U}(1.0, 1.5)$. The bounding box size is set as 1.0. For large scene generation, we enlarge the range of camera radius to $\mathcal{U}(0.1, 2.3)$ for better details

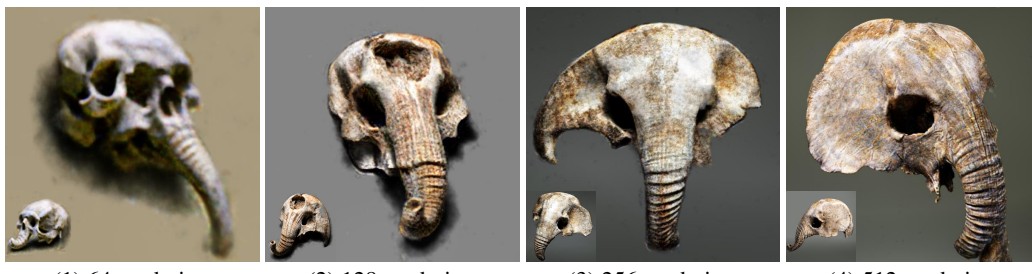

| (1) 64 rendering. | (2) 128 rendering. | (3) 256 rendering. | (4) 512 rendering. |

Figure 14: Ablation study on rendering resolution during NeRF training. Training with higher resolution produces better results with finer details. However, our VSD still provides competitive results under a lower resolution (128 or 256), which is more computationally efficient. The prompt here is *an elephant skull*.

and geometry consistency. We enlarge the bounding box size to $5.0$. For large scenes with a centric object, we set the range of camera radius as $\mathcal{U}(1.0, 2.0)$.

DreamFusion [34] introduces an explicit shading model based on normal vectors during the training process to enhance the geometry. We currently disable the shadings and it reduces the computational cost and memory consumption. We leave incorporating the shading model into our method as future work.

**Optimization.** We optimize $25k$ steps for each particle with AdamW [26] optimizer in NeRF stage. We optimize $15k$ steps for geometry fine-tuning and $30k$ steps for texture fine-tuning. At each stage, for the first $5k$ steps we sample time steps $t \sim \mathcal{U}(0.02, 0.98)$ and then directly change into $t \sim \mathcal{U}(0.02, 0.50)$. For large scene generation, we delay the annealing time to $10k$ steps since large scene generation requires more iterations to converge. The NeRF training stage consumes $17/17/18/27$GB GPU memory with $64/128/256/512$ rendering resolution and batch size of $1$. The Mesh fine-tuning stage consumes around $17$GB GPU memory with $512$ rendering resolution and batch size of $1$. The whole optimization process takes around several hours per particle on a single NVIDIA A100 GPU. We believe adding more GPUs in parallel will accelerate the generation process, and we leave it for future work.

**Licenses** Here we provide the URL, citations and licenses of the open-sourced assets we use in this work.

Table 2: URL, citations and licenses of the open-sourced assets we use in this work.

| URL | Citation | License |
|---|---|---|
| https://github.com/ashawkey/stable-dreamfusion | [51] | Apache License 2.0 |
| https://github.com/threestudio-project/threestudio | [13] | Apache License 2.0 |
| https://github.com/Stability-AI/stablediffusion | [38] | MIT License |
| https://github.com/NVIDIAGameWorks/kaolin | [11] | Apache License 2.0 |
| https://github.com/huggingface/diffusers | [54] | Apache License 2.0 |

## G  2D Experiments of Variational Score Distillation

Here we describe the details of the experiments on 2D images with Variation Score Distillation in the main text of Fig. 3. Here we set the number of particles as $8$. We train a smaller U-Net from scratch to estimate the variational score since the distribution of several 2D images is much simpler than the distribution of images rendered from different poses of 3D structures. The optimization takes around $6000$ steps. Additional U-Net is optimized $1$ step on every optimization step of the particle images. The learning rate of particle images is $0.03$ and the learning rate of U-Net is $0.0001$. The batch size of particles and U-Net are both set as $8$. Here we do not use annealed time schedule for both VSD and SDS for a fair comparison.

To further demonstrate the effectiveness of our VSD, we increase the number of particles to a large number of 2048. The optimization takes around $20k$ steps. The batch size of U-Net is 192 and the batch size of particles is set as 16. The results are shown in Figure 15. It can be seen from the figure that our VSD generates plausible results even under a large number of particles, which demonstrates the scalability of VSD. Although, due to the high optimization cost of 3D representation, the number of particles in 3D experiments is relatively small, we demonstrate that VSD has the potential to scale up to more particles.

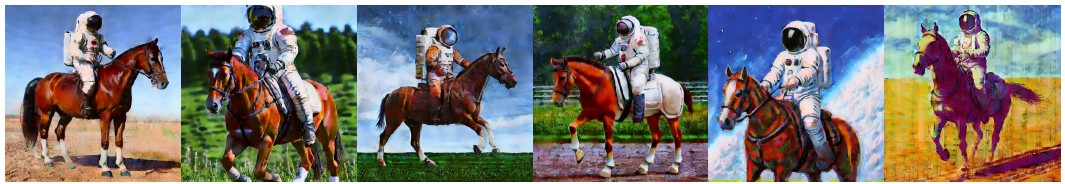

Figure 15: Selected samples from 2048 particles in a 2D experiment of VSD. Our VSD generates plausible results even under a large number of particles, which demonstrates the scalability of VSD. The prompt is *an astronaut is riding a horse.*

To match the 3D experiment setting, we also provide 2D experiments with LoRA and add annealed time schedule. The results are shown in Figure 16. The number of particles is set as 6 and CFG= 7.5. Our VSD provides high-fidelity and diverse results in this setting. We also set the number of particles as 2048 and the results are shown in Figure 17.

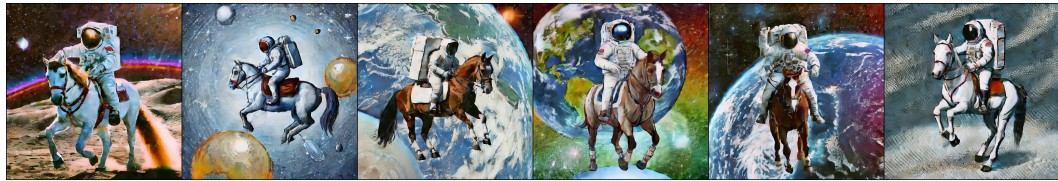

Figure 16: 2D experiments of VSD (CFG= 7.5) results with LoRA and annealed time schedule. The number of particles is 6. Our VSD provides high-fidelity and diverse results in this setting. The prompt is *an astronaut is riding a horse.*

We also provide the results of SDS for comparison in Fig. 18. Compared to our VSD, the results of SDS are smoother and lack details.

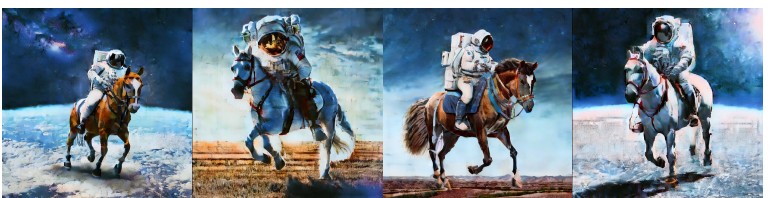 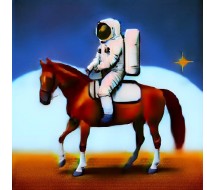

Figure 17: Selected samples of 2D experiments of VSD (CFG= 7.5) results with LoRA. The number of particles is 2048.

Figure 18: SDS result in 2D.

Moreover, we visualize VSD/SDS training phase of 2D in Fig. 19. Since the gradient is not directly readable, we visualize $x + \Delta x$, which is the updated results if current sample optimizes via this gradient direction. As shown in Fig. 19, SDS tends to provide over-saturated and over-smooth gradient while VSD provides more natural-looking gradients with more details. As a consequence, VSD provides better final results.

## H How will CFG weights affect diversity?

In this section, we explore how CFG affects the diversity of generated results. For VSD, we set the number of particles as 4 and run experiments with different CFG. For SDS, we run 4 times

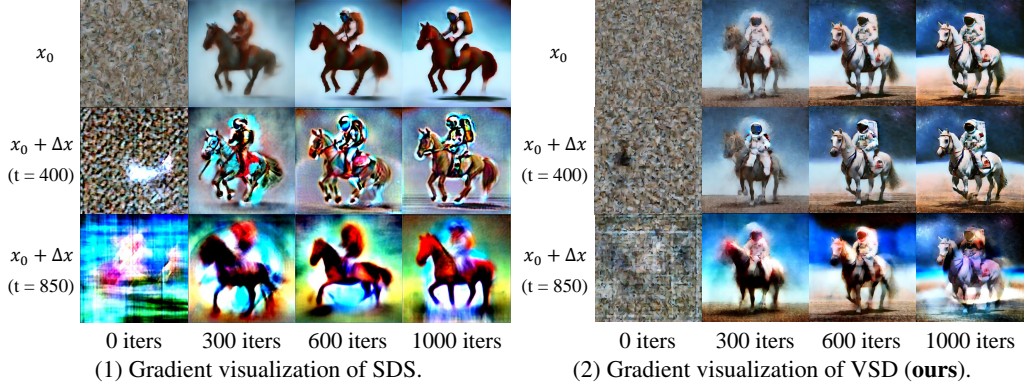

|  | 0 iters | 300 iters | 600 iters | 1000 iters |  | 0 iters | 300 iters | 600 iters | 1000 iters |

(1) Gradient visualization of SDS.  (2) Gradient visualization of VSD (**ours**).

Figure 19: Gradient visualization of VSD and SDS during the training phase in 2D experiment. SDS tends to provide over-saturated and over-smooth gradient while VSD provides more natural-looking gradients with more details.

of generation with different random seeds. The results are shown in Figure 20. As shown in the figure, smaller CFG provides more diversity. We conjecture that this is because the distribution of smaller guidance weights has more diverse modes. However, when the CFG becomes too small (e.g., CFG= 1), it cannot provide enough guidance strength to generate plausible results. Therefore, for all our 3D experiments shown in the results, we set CFG = 7.5 as a trade-off between diversity and optimization stability. Note that SDS could not work well in such small CFG weights. Instead, our VSD provides a trade-off option between CFG weight and diversity, and it can generate more diverse results by simply setting a smaller CFG.

## I  Limitations and Discussions

Although ProlificDreamer achieves remarkable text-to-3D results, the generation takes hours (especially for high-resolution NeRF training), which is much slower than the vanilla image generation by a diffusion model. Speeding up the text-to-3D generation is another critical problem, and we leave it in future work.

Besides, our proposed scene initialization demonstrates its effectiveness in generating expansive scenes, yet there are still limitations, particularly with respect to camera positioning. Our current model sets the camera with a fixed view toward the scene's center, which serves object-centric scenes effectively but may be suboptimal for scenes with intricate geometry and detailed textures. Furthermore, despite our efforts to produce outcomes with a rich structure, occasional failures occur and the geometry reverts to a simplistic textured sphere. In order to address these limitations, future research could focus on developing improved camera poses capable of capturing and rendering scenes in finer detail. Furthermore, our present model relies solely on the text-to-image diffusion model without the assistance of other models. The integration of off-the-shelf depth estimation models [37], as utilized in other studies [16, 58], could potentially enhance the accuracy and detail of the scene generation process.

In addition, the correspondence between text prompts and generated results is sometimes insufficient, especially for complex prompts. We conjecture that it is because the ability to generate from complex prompts is limited by the text encoder of Stable Diffusion. In addition, the multi-face Janus problem [57] also exists in our case. Nevertheless, we believe our proposed VSD and other contributions are orthogonal to these problems, and the generation quality can be further improved by introducing more techniques, such as using a more powerful text-to-image diffusion model that understands view-dependent prompts better [34, 41], or a diffusion model with more 3D priors [24].

## J  User Study

For completeness, we follow previous works [20, 4] and conduct a user study by comparing ProlificDreamer with DreamFusion [34], Magic3D [20] and Fantasia3D [4] under 15 prompts, 5 prompts for each baseline. Since none of the baselines have released their codes, we can only use

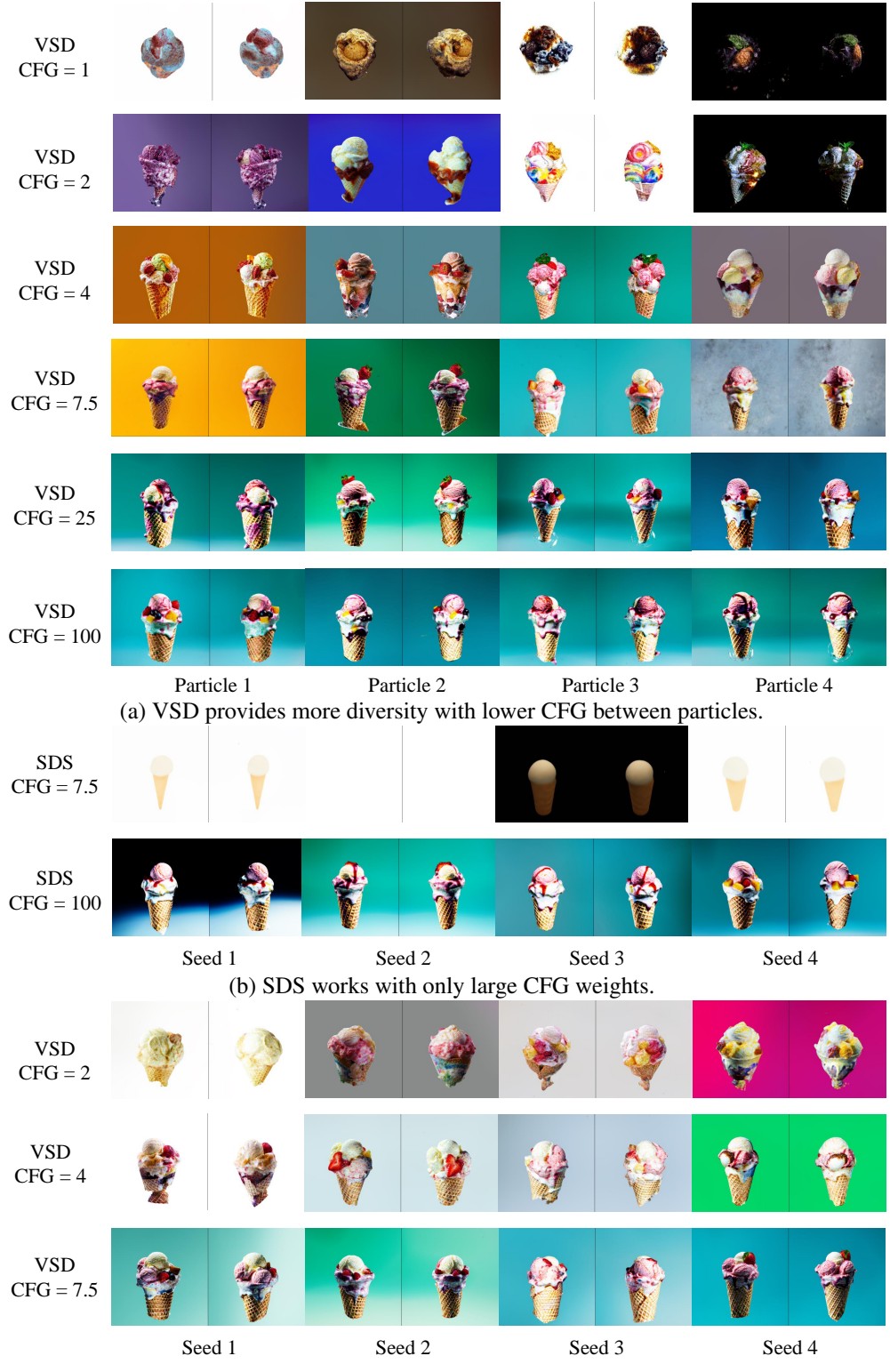

(a) VSD provides more diversity with lower CFG between particles.

(b) SDS works with only large CFG weights.

(c) VSD with single particle and different seeds provide slightly less diversity.

Figure 20: Ablation on how CFG weight affects the randomness. Smaller CFG provides more diversity. But too small CFG provides less optimization stability. The prompt is *A high quality photo of an ice cream sundae*.

the figures from their papers, which limits the number of results from baselines for us to compare. So we currently only compare under 15 prompts. The volunteers are shown the generated results of our ProlificDreamer and baselines and asked to choose the better one in terms of fidelity, details and vividness. We collect results from 109 volunteers, yielding 1635 pairwise comparisons. The results are shown in Table 3. Our method outperforms all of the baselines.

Table 3: Results of user study. The percentage of user preference ($\uparrow$) is reported in the table.

| Name | DreamFusion | Magic3D | Fantasia3D |
|---|---|---|---|
| Prefer baseline | 6.87 | 5.50 | 9.73 |
| Prefer ProlificDreamer (Ours) | **94.13** | **94.50** | **90.27** |

# K  Quantitative Results

Table 4: 3D sample quality by SDS or VSD, 100 prompts.

| Method | SDS | VSD ($n$=1) |
|---|---|---|
| 3D-FID ($\downarrow$) | 118.92 | 107.02 |

Table 5: 3D sample quality by SDS or VSD, 25 prompts.

| Method | SDS | VSD ($n$=1) | VSD ($n$=4) |
|---|---|---|---|
| 3D-FID ($\downarrow$) | 191.82 | 186.87 | 185.88 |

In this section, we add some quantitative results to demonstrate the effectiveness of VSD.

**Experiment Setting**   For 3D experiments, we compute the FID score between rendered images by SDS/VSD and 2D sampled images by ancestral sampling, named as 3D-FID. Specifically, we select 100 prompts from previous works including DreamFusion, Magic3D and Fantasia3D. For each prompt, we use VSD (number of particles $n$=1, CFG=7.5) or SDS (CFG=100) to optimize one 3D object and render 10 images uniformly from the circumference at an angle of 30° above the horizon, and collect 1k images in total. To isolatedly compare VSD with SDS, we run with the default setting of the stage-1 NeRF training of ProlificDreamer (i.e., both VSD and SDS are in 512 resolution and use annealed $t$).

In Table 4, We compute the FID score between the 1k samples from the 100 prompts and a 50k reference batch, which is sampled by 50-step DPM-Solver++ [28] with 500 images per prompt. In Table 5, due to the time and computation resource limits, we compare the results for VSD with $n$=4 under only 25 randomly-selected prompts from the aforementioned 100 prompts, and compare SDS, VSD ($n$=1) and VSD ($n$=4) with the 50-step DPM-Solver++ under these 25 prompts. For VSD ($n$=4), as we can get 4 particles (3D objects) per prompt, we randomly select one particle per prompt and render the corresponding 10 images of the selected particle for fair comparison. For 2D experiments in Table 6, we follow the common setting of evaluating text-to-image models by computing FID on MSCOCO2014 validation set. Specifically, we randomly select 1k prompts and sample one image per prompt by either 50-step DPM-Solver++, SDS (CFG=100), VSD ($n$=4, CFG=7.5) or VSD ($n$=8, CFG=7.5) to collect 1k samples for each method, and then compute the FID between the obtained samples and the entire COCO validation set. For VSD, as we can get $n$ images per prompt, we randomly select one image per prompt for fair comparison.

**Results**   VSD with $n$=1 still outperforms SDS in 3D (both with 512 resolution and annealed $t$), as shown in Table 4 and Table 5, which demonstrates the effectiveness of VSD.

Using more particles is slightly better. Due to the limitation of time and computation resources, we only compare $n$=1 and $n$=4 in 3D experiments, and $n$=4 with $n$=8 in the 2D experiments. As shown in Table 5, VSD with 4 particles slightly outperforms VSD with 1 particles in the 3D setting; and as shown in Table 6, VSD with 8 particles slightly outperforms VSD with 4 particles in the 2D setting.

Table 6: 2D sample quality by different samplers, 1000 prompts.

| Method | SDS | VSD (n=4) | VSD (n=8) | DPM++ |
|---|---|---|---|---|
| FID ($\downarrow$) | 90.09 | 68.02 | 66.68 | 47.91 |

VSD outperforms SDS in 2D. As shown in Table 6, the FID by VSD is much better than SDS. As the 2D setting isolates the sampling algorithm from the 3D representations, we can directly compare different sampling algorithms, finding that VSD can get better sample quality than SDS (though still worse than SOTA diffusion samplers, it can generalize to 3D cases).

## L  Why using SDS in stage-2 for the geometry optimization of mesh?

VSD can also be used to generate geometry. To validate this, we provide an ablation example in Fig. 21 (3a),(3b). As shown in the figure, VSD can obtain reasonable geometry. Although the some part of the geometry from VSD is with more details than SDS (including the tail of the horse), on the whole, the result from VSD is similar with SDS. We conjecture that this is because currently the triangle size of the mesh is relatively large and can't represent very fine details. Thus, for efficiency, we use SDS instead of VSD for mesh geometry optimization. We believe that VSD can be used to obtain high quality mesh if more advanced mesh representation is available.

Moreover, despite that we use SDS to optimize the geometry in stage-2, VSD is still crucial in stage-1 and stage-3, in which VSD significantly improves the generated quality.

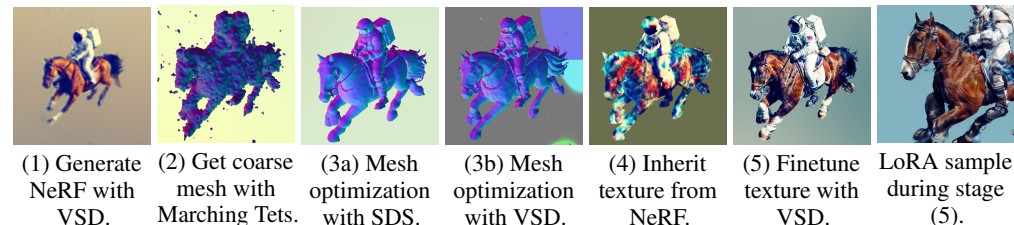

(1) Generate NeRF with VSD. (2) Get coarse mesh with Marching Tets. (3a) Mesh optimization with SDS. (3b) Mesh optimization with VSD. (4) Inherit texture from NeRF. (5) Finetune texture with VSD. LoRA sample during stage (5).

Figure 21: Pipeline along with ablation study of VSD for geometry optimization. In (3a) and (3b), we show that VSD can also be used for mesh optimization of geometry. We also show intermediate results of our pipeline in this figure along with the samples from LoRA during the training phase in (5).

## M  More Comparisons with Baselines

Here, we provide more comparisons with baselines in Fig. 22, Fig. 23, Fig. 24, Fig. 25, Fig. 26 and Fig. 27. Since none of the baselines have released their codes, we can only directly copy the figures from the corresponding papers. Some baselines are missing given a specific prompt because the prompt is not included in the corresponding papers. To demonstrate geometry, some baselines choose textureless shading [34, 20], while the other [4] prefers the normal map. For ProlificDreamer, we uniformly show the normal map for consistency. As shown in the figures, our method achieves better results in terms of fidelity and details.

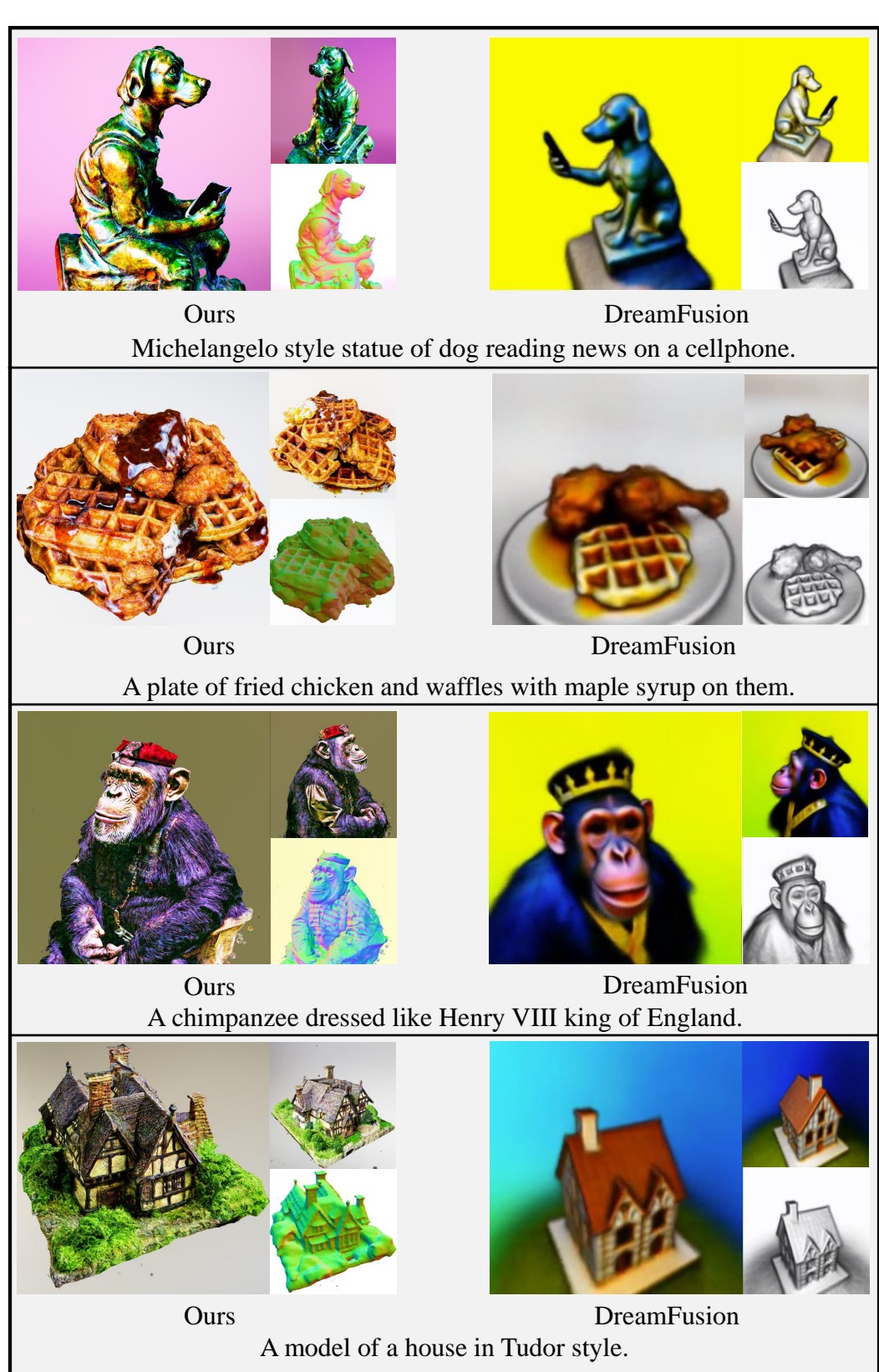

Figure 22: More results of ProlificDreamer compared with baselines.

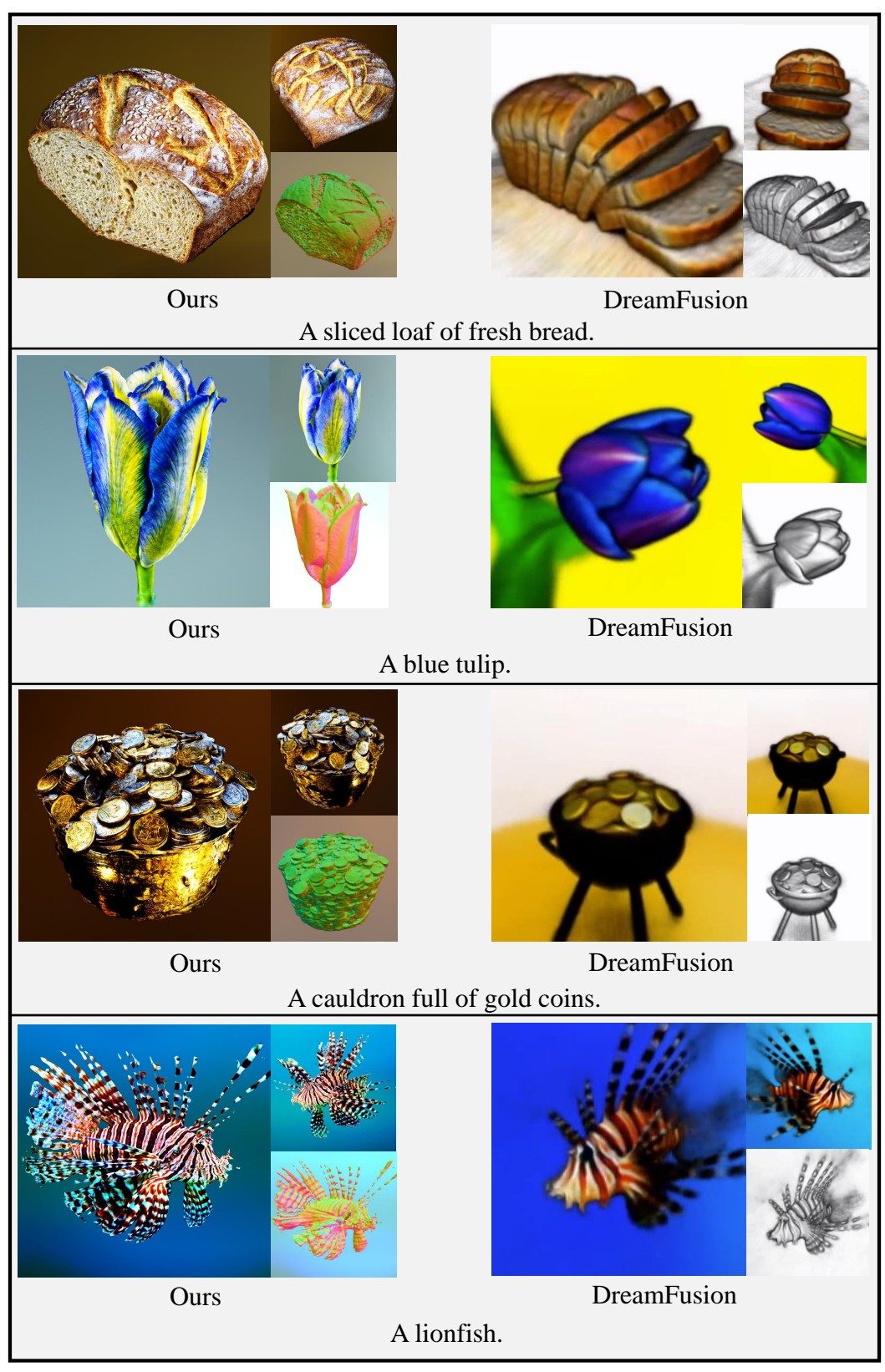

Figure 23: More results of ProlificDreamer compared with baselines.

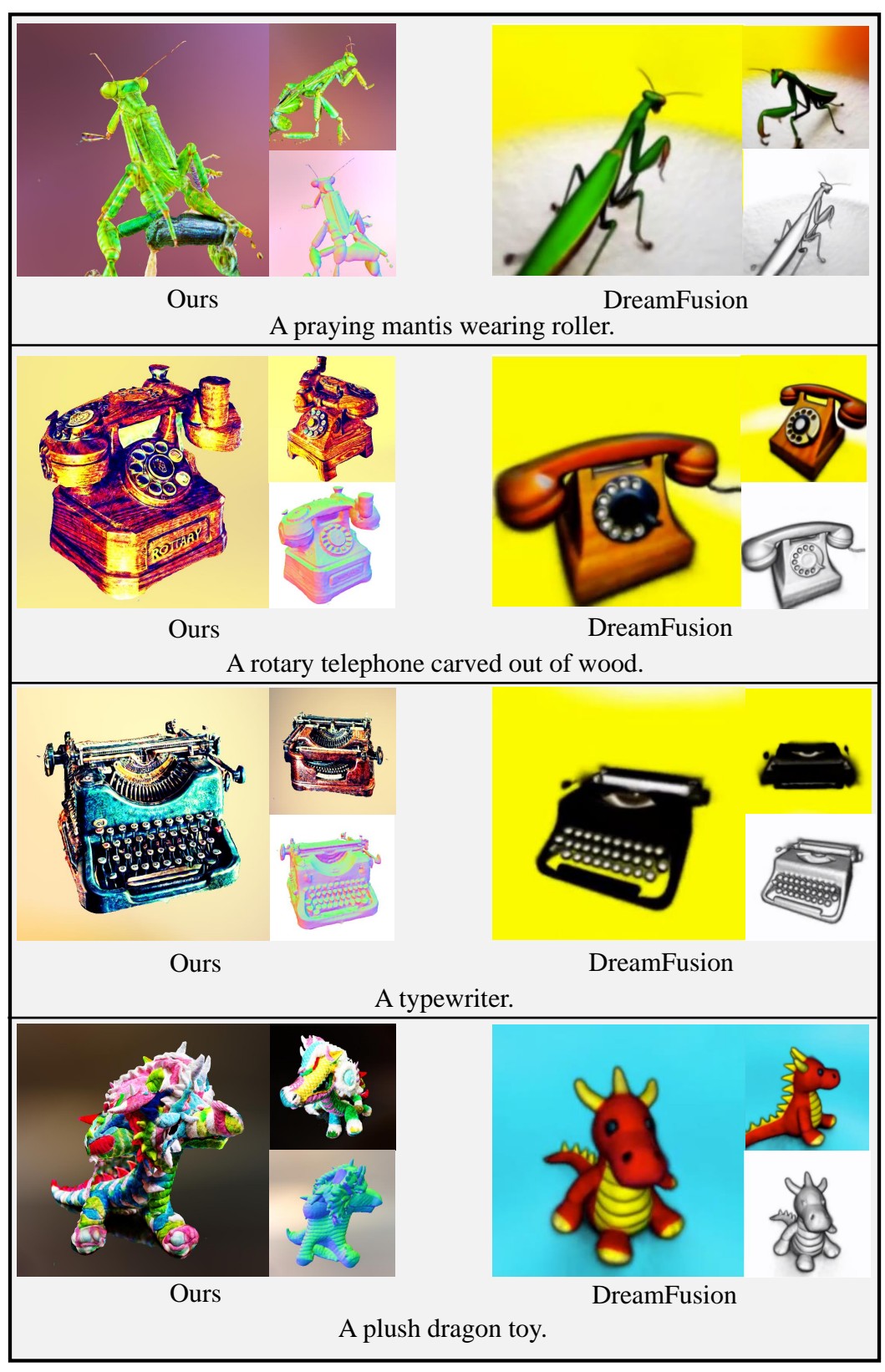

Figure 24: More results of ProlificDreamer compared with baselines.

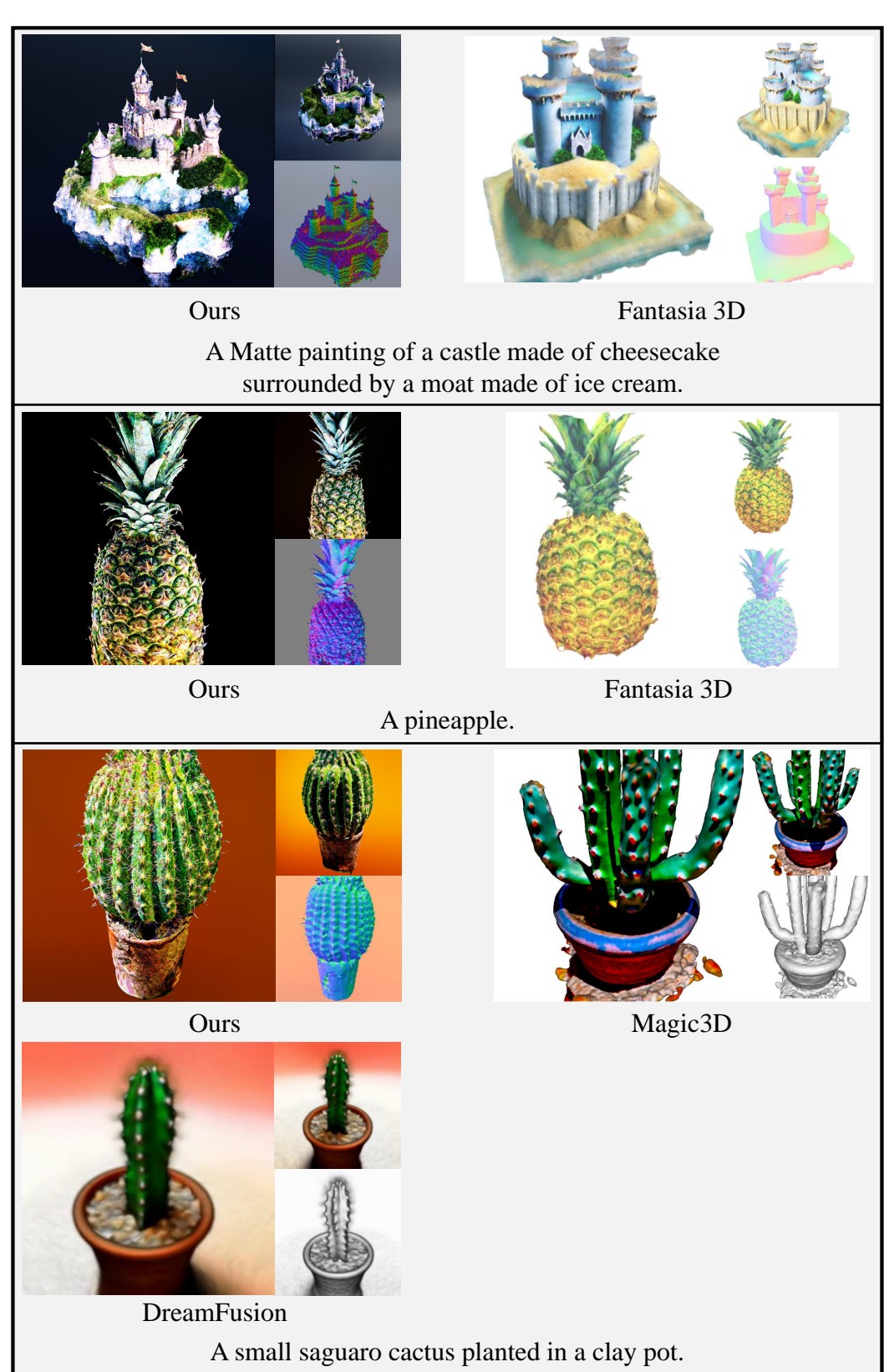

Figure 25: More results of ProlificDreamer compared with baselines.

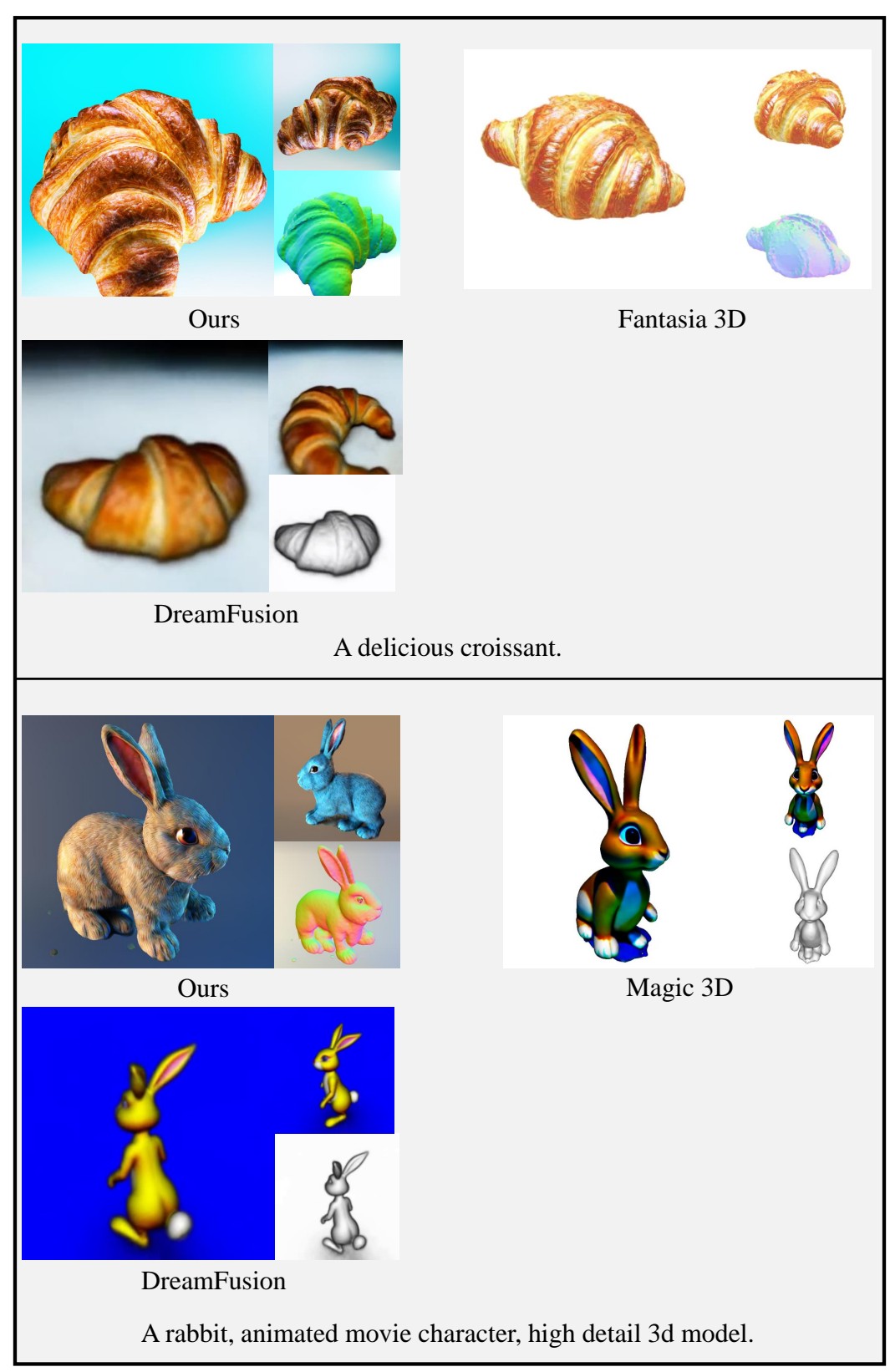

Figure 26: More results of ProlificDreamer compared with baselines.

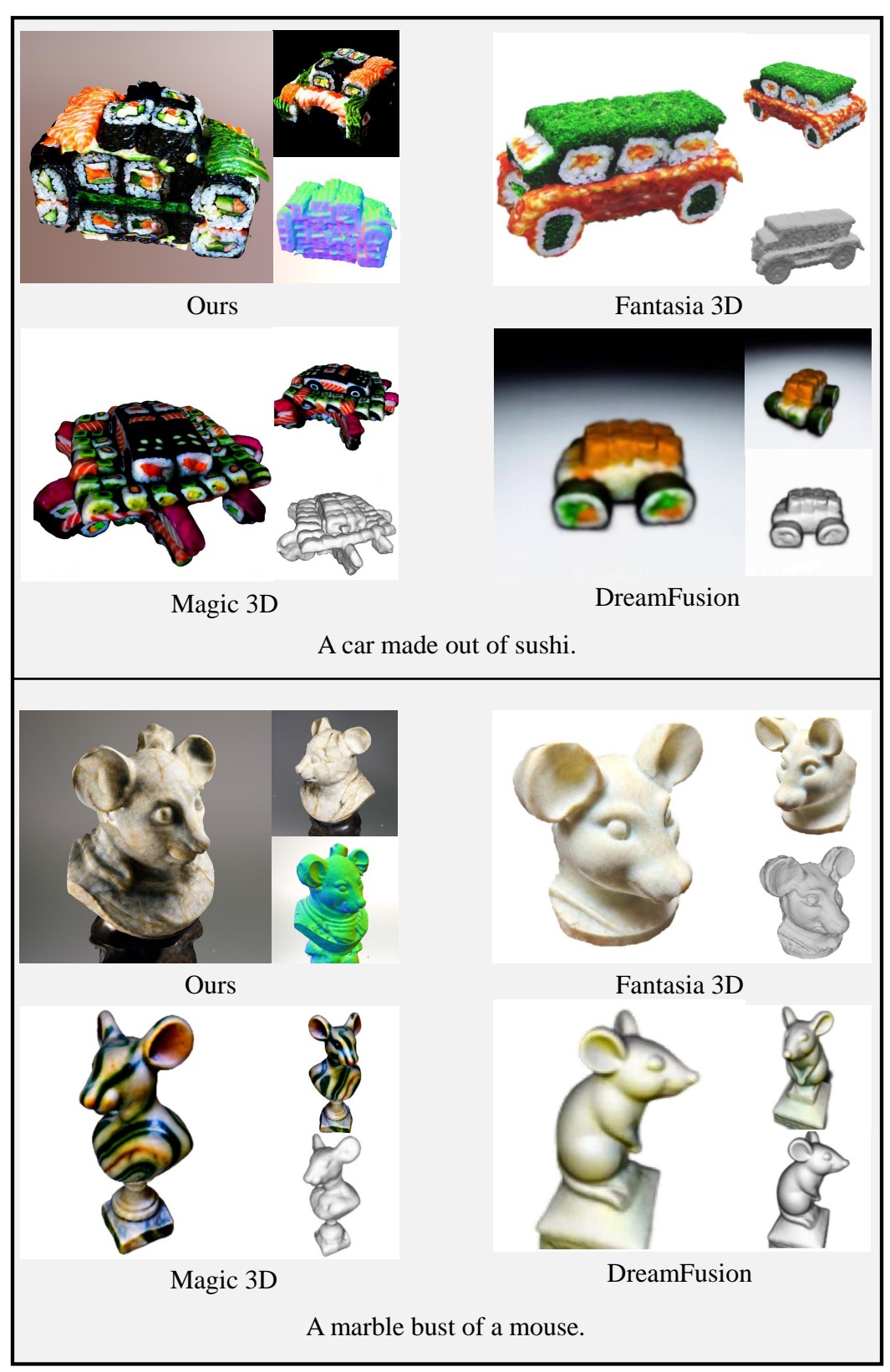

Figure 27: More results of ProlificDreamer compared with baselines.

