# OpenReview forum: "ProlificDreamer: High-Fidelity and Diverse Text-to-3D Generation with Variational Score Distillation"
_NeurIPS.cc/2023/Conference — NeurIPS 2023 spotlight_

### Official Review · Reviewer_eq7N · 2023-06-29

**Soundness:** 3 good
**Presentation:** 2 fair
**Contribution:** 3 good
**Rating:** 7
**Confidence:** 5

**Summary:**

 The paper presents a nice distillation pipeline that can perform high-quality text-to-3D generation using 2D diffusion priors. By introducing the score on NeRF / mesh rendered images, the authors make score distillation work with a classifier-free guidance weight as low as 7.5. This approach largely improves the detail quality of the converged results and inspires the community with an exciting direction.

**Strengths:**

- The authors propose to estimate scores on rendered image with a LoRA framework fine-tuned for the target scene.
- By introducing this additional score, the distillation framework is able to work with low classifier-free guidance weight at 7.5, yielding much better appearance than the original SDS loss in DreamFusion.
- Both 2D and 3D experiments validate the expressiveness of the proposed framework.

**Weaknesses:**

- The paper starts with derivations to support the particle-based variational framework. However, using more particles doesn't seem to help with the generation results. Is using a single particle not enough? What's the benefit of multiple particles, and when do we need them?
- In Eq.(8) for 3D experiments, the authors mention that there's a `c` for camera pose added for the conditioning of the diffusion model. How is this implemented? Is this camera pose necessary for the framework to work? It would be nice if there's an ablation study on removing the camera pose conditioning.
- For 3D experiments, why do we need a LoRA model fine-tuned from a pre-trained Stable Diffusion? Does using a small U-Net work for 3D case?
- For the mesh refinement stage, the authors follow Fantasia3D to refine the geometry and texture separately. SDS loss is used in the geometry stage that refines normal maps. I'm interested in whether the proposed vsd loss is helpful in this case. Adding a comparison experiment that uses VSD to refine the normal map will be great.
- Existing methods mostly use view-dependent prompts, meaning that text embedding `y` and camera `c` are not independent. Do authors use view-dependent prompts in the experiments?
- Why are the two rows in Eq. (5) equal? More derivations are appreciated.
- Is v-prediction necessary for the LoRA model? Does the proposed framework work with eps prediction?


**Questions:**

Please refer to the weakness section.

**Limitations:**

Please refer to the weakness section.

---

> ### Author Rebuttal · Authors · 2023-08-10
>
> Thank you for your valuable review and suggestions. Below we provide a point-to-point response to all comments. If our response has addressed the concerns and brings new insights to the reviewer, we will highly appreciate it if the reviewer considers raising the score.
>
> ***Q1: Is using a single particle not enough? What's the benefit of multiple particles, and when do we need them?***
>
> **A**: We find that using a single particle is **enough** for 3D generation, since the LoRA model has a strong image prior and performs quite well in the few-shot learning scenario. Specifically, our demonstrated results in Fig.1(a-b) and Appendix A are all obtained by optmizing a single particle with our proposed pipeline. As for the benefits of multiple particles, please refer to the ***common response, Q3***. In general, increasing the number of particles brings more diversity and slightly increases the generated quality, and we need them when we want more diverse and higher-quality results. We will add a corresponding discussion in the final version to improve the writing.
>
>
> ***Q2: How is the camera condition implemented in the variational model? Is this camera pose necessary for the framework to work? It would be nice if there's an ablation study on removing the camera pose conditioning.***
>
> **A**: As presented in our appendix G, the camera pose is fed into a 2-layer MLP and then added into the time step embeddings of the base model. This camera condition is beneficial for the algorithm to converge since it makes LoRA training easier. We provide an ablation study on this in the ***response pdf, Fig.4***, showing that the results without camera conditions are inferior to the results with camera conditions, and the results without camera condition tend to have floaters and degraded geometry.
>
>
> ***Q3: For 3D experiments, why do we need a LoRA model fine-tuned from a pre-trained Stable Diffusion? Does using a small U-Net work for 3D case?***
>
> **A**: Due to the computation cost, the number of particles is limited to a small number (1~4) in 3D experiments. Thus, estimating the score functions of noisy rendered images is a **few-shot learning problem** of diffusion models. As LoRA has a strong image prior and is suitable for this few-shot learning setting, we find that using LoRA can greatly improve the sample quality by VSD. In addition, in our early experiments, we have tried using a small U-Net instead of LoRA, and we find that the samples have more artifacts than those of LoRA.
>
>
> ***Q4: Is the proposed vsd loss is helpful in stage2 (mesh geometry finetuning)? Adding a comparison experiment that uses VSD to refine the normal map will be great.***
>
> **A**: Please refer to **common response, Q2**.
>
>
> ***Q5: Existing methods mostly use view-dependent prompts, meaning that text embedding $y$ and camera $c$ are not independent. Do authors use view-dependent prompts in the experiments?***
>
> **A**: We indeed use view-dependent prompts for pretrained Stable Diffusion, following previous works such as Dreamfusion and Magic3D. However, we do not use view-dependent prompts for the variational model (i.e., the LoRA model) since the camera pose has been injected into the model explicitly. We will change the math notations correspondingly to align them with the settings of view-dependent prompts, as specified in the following response.
>
>
> ***Q6: Why are the two rows in Eq. (5) equal? More derivations are appreciated.***
>
> **A**: Thank you for pointing it out. We appologize for abusing the notation $y$ in the submission. In fact, the $y$ appears in the pretrained distribusion $p_t(x_t|y)$ and model $\epsilon_{\text{pretrain}}(x_t,t,y)$ depends on the camera condition $c$. We will make the dependence explicit by using notation $y^c$. Then Eq.(5) naturally holds (actually the first line is redundant). We will fix it in the final version and note that the method and theorems are still sound and the implementation remains the same.
>
> ***Q7: Is v-prediction necessary for the LoRA model? Does the proposed framework work with eps prediction?***
>
> **A**: In our early experiments, we find that eps-prediction for the LoRA model can also work, but the performance of eps-prediction is slightly inferior to v-prediction in terms of texture details. We also provide ablation study on this in ***response pdf, Fig.4*** to demonstrate this.
>
> Once again, thank you for your constructive feedback and for considering our paper for acceptance. We'll revise our paper according to your suggestions. We kindly request that you consider raising the score accordingly if we addressed your concerns.

---

### Official Review · Reviewer_Gck3 · 2023-07-03

**Soundness:** 4 excellent
**Presentation:** 3 good
**Contribution:** 4 excellent
**Rating:** 7
**Confidence:** 3

**Summary:**

This paper proposes an interesting and novel technique for the task of text-to-3D generation. It defines a distribution of the target 3D scene, which is implemented as particles. Given the textual description, the distribution is updated using the Wasserstein gradient flow. Moreover, the paper proposes to fine-tune a diffusion model based on LoRA to calculate the score function of noisy rendered images, which is used to compute the objective function. Experiments show that the approach significantly improve the generation quality.

**Strengths:**

1. The proposed technique is novel and interesting. Modeling 3D scenes as a distribution is insightful, which could inspire the following research.
2. It is a good idea to fine-tune LoRA-based diffusion models on noisy rendered images for calculating the objective function.
3. The results of the proposed model are impressive and obviously outperform previous methods.

**Weaknesses:**

1. I am curious about how the number of particles affect the performance of the proposed model.
2. Can this idea be used in amortized text-to-3D generative models? Does it consumes a lot of GPU memory?
3. It would be better to perform ablation studies on the distillation time schedule and the density initialization.

**Questions:**

Please see the weakness section.

**Limitations:**

The authors have discussed the limitations.

---

> ### Author Rebuttal · Authors · 2023-08-10
>
> Thank you for your valuable review and suggestions. Below we provide a point-to-point response to all comments. If our response has addressed the concerns and brings new insights to the reviewer, we will highly appreciate it if the reviewer considers raising the score.
>
> ***Q1: I am curious about how the number of particles affect the performance of the proposed model.***
>
> **A**: Please refer to the **common response, Q3**.
>
> ***Q2: Can this idea be used in amortized text-to-3D generative models? Does it consumes a lot of GPU memory?***
>
> **A**: Yes, it is quite interesting and promising to use our method for amortized text-to-3D generative model, and we leave it for future work. Currently, our method consumes 27GB GPU memory in NeRF training stage and 17GB in mesh finetuning stage, using stable-diffusion 2.1 and its corresponding LoRA model.
>
> ***Q3: It would be better to perform ablation studies on the distillation time schedule and the density initialization.***
>
> **A**: We have provide ablation studies on the distillation time schedule in Figure 5 in our paper, which shows that the generated results with our proposed annealed-time-schedule method have more details. As for density initialization, we find that without scene initialization, scene-level generation did not work at all, so we did not post an image on this in our paper. We will add a corresponding discussion in the final version.
>
> Once again, thank you for your constructive feedback and for considering our paper for acceptance. We'll revise our paper according to your suggestions. We kindly request that you consider raising the score accordingly if we addressed your concerns.

---

### Official Review · Reviewer_krEH · 2023-07-03

**Soundness:** 3 good
**Presentation:** 3 good
**Contribution:** 4 excellent
**Rating:** 7
**Confidence:** 4

**Summary:**

In this paper, a novel method for generating 3D representations from text prompts is proposed. More specifically, a novel variational score distillation (VSD) based optimisation strategy is proposed that allows for optimising NeRF- and mesh-based 3D representations by utilising pre-trained 2D diffusion models. It improves over the previously in Dreamfusion proposed score distillation sampling (SDS) by optimising a distribution over the 3D parameters instead of a single point Dirac distribution. Further, the authors made additional engineering improvements to the text-to-3d pipeline. The proposed system does not rely on high CFG weights as required for SDS and leads to impressive results that compare favourably against the state-of-the-art.


**Strengths:**

- The authors clearly identify an important shortcoming of current text-to-3D approaches: suffering from over-saturation, over-smoothing, and low-diversity problems. They identify the SDS-based optimisation as the core problem, and propose a theoretically-sound alternative that can be trained with only a small overhead compared to SDS.
- The shown results clearly look impressive and the obtained scene representations clearly improve over the state-of-the-art results.
- The manuscript is (mostly) very well written, has a great structure, and a great "reading flow".
- The performed experiments, including the 2D experiments, clearly highlight the authors' contributions and show the effectiveness of the individual components of the proposed methods.
- The authors provide a clear and correct overview over their core contributions in Table 1 and demonstrate their familiarity with the fast-moving field of text-to-3D generation.


**Weaknesses:**

- Number of particles: I am sceptical if n=1 or n=4 (L. 150) is really enough to capture the distribution. It would be interesting to investigate the relationship of number of particles and the quality of the results; as it seems that the prediction network for the noisy rendered images is trained with all particles, it could be that the quality overall improves with more samples.

- Geometry optimisation and VSD/SDS: Accordingly to L. 247 ff., I believe that the authors explain that the mesh geometry is fine-tuned with SDS, not VSD (see also question below). If understood correctly, this part should express more clearly that it is only referring to the mesh geometry, not the overall geometry optimisation. Further, this seems to be an interesting limitation of the current method. It would be interesting to see a respective ablation figure comparing SDS and VSD for this mesh-geometry prediction stage.

- Geometry Diversity: Figure 1 c.) shows the diversity of samples, and it appears that the geometry diversity is limited compared to the texture diversity. It would be interesting to discuss potential reasons for this.

- Experimental Results: No quantitive metrics are reported and the only user study is "hidden" in Table 3 of the supplementary material. While it is true that the task of text-to-3D is very challenging to be measured quantitatively, metrics reported by previous methods such as Dreamfusion could also be reported, and the user study can be shown in the main paper or at least be referenced.

- Camera Pose Priors: It would be very interesting to see how the camera sampling strategies affect the results; in particular for the "scene generation" experiments. Have the authors started investigating this dimension of the problem?

- Related work: A good and thorough related work discussion is contained only in the supplementary. I would encourage the authors to add a discussion, if accepted, to the main manuscript.

- Noisy prediction network: The scores for the noisy real and the noisy rendered images are predicted by two separate models (L. 166), and the one for the rendered images is smaller / a low-rank approximation. Is this introducing some form of imbalance or is this not a problem for the optimisation?

- Training of noisy renderings score prediction network: If understood correctly from the supplementary, the network for predicting the scores of the noisy rendered images is trained simultaneously with the NeRF model. It could be interesting to study how different optimisation schemes affect the results.

- Input text prompt: I believe it is not stated whether the same text prompt or different text prompts are used for different camera poses.

Typos:
- L 178: speical case -> special case
- L 283: variation score distillation -> variational score distillation
- L. 284: 3D parameter -> 3D parameters
- L 637 of supp mat: an addition diffusion model -> an additional diffusion model

**Questions:**

- Have the authors investigated whether there is a relationship between number of particles and the quality of the results?
- Could the authors expand on the limited geometry variance in contrast to the texture variance, and why the use of the VSD is not improving over SDS for the mesh geometry prediction?
- Could the authors expand on the importance of used camera pose priors, especially in the context of scene generation?
- Are different text prompts used for different camera poses?

**Limitations:**

The authors discuss limitations both in the main paper as well as more extensively in the supplementary. I believe the discussion is thorough and good as is.

---

> ### Author Rebuttal · Authors · 2023-08-10
>
> Thank you for your valuable review and suggestions. Below we provide a point-to-point response to all comments. If our response has addressed the concerns and brings new insights to the reviewer, we will highly appreciate it if the reviewer considers raising the score.
>
> ***Q1: Is n=1~4 is really enough to capture the distribution? The relationship of number of particles and the quality?***
>
> **A**: Yes, it is enough because we adopt LoRA for the variational distribution. LoRA has a strong image prior and is suitable for few-shot score function learning. We have dicussed and provided corresponding quantitative experiments in ***common response, Q3***. In general, increasing the number of particles slightly increases the generated quality.
>
> ***Q2: Why the use of the VSD is not improving over SDS for the mesh geometry prediction?***
>
> **A**: Please refer to **common response, Q2** for the discussion on this topic. We have provided a respective ablation figure comparing SDS and VSD for this mesh-geometry prediction stage, please see **response pdf, Fig.5**.
>
> ***Q3: Discuss potential reasons why the geometry diversity is limited compared to the texture diversity (Fig.1.c).***
>
> **A**: The samples by VSD also have diverse geometry for some other prompts, and we provide extra results which demonstrates more geometry diversity in the ***response pdf, Fig.5***. We conjecture that the demonstrated two prompts in Fig.1.c are limited in geometry diversity, due to the image prior in stable-diffusion. In addition, lower the cfg will bring more geometry diversity as shown in Figure 17 in our appendix.
>
>
> ***Q4: Quantitative evaluation results?***
>
> **A**: We provide quantitative results in **common response, Q1**, including both 2D experiments (evaluated on 1000 prompts) and 3D experiments (evaluated on 100 prompts). It will be added into the main paper. We will also reference the user study in the main paper.
>
> ***Q5: Add related work to the main manuscript.***
>
> **A**: Sure, we will add the related work into the main text.
>
>
> ***Q6: Is the smaller / low-rank approximation model for the noisy rendered images introducing some form of imbalance?***
>
> **A**: No, the training procedure by using LoRA is stable and we did not notice any problems for the optimization. We believe that it is because LoRA has a strong image prior and is suitable for few-shot learning (estimating the score functions of noisy rendered images), so the estimated score (update direction) in VSD is meaningful (please see ***response pdf, Fig.2*** for visualization of the updating direction of VSD). Moreover, LoRA only trains a small amount of parameters, and the base architecture is the same as the pretrained stable-diffusion, so the capacity of these two models is balanced.
>
> ***Q7: It could be interesting to study how different optimization schemes for the noisy renderings score prediction network affect the results.***
>
> **A**: During the early experiments, we have tried different training schemes and find that optimizing 3D particles and variational scores for one step alternatively performs the best.
>
> ***Q8: Typos***
>
> **A**: Thanks for correcting our typos and helping to improve the writing quality. We will fix them in the final version correspondingly.
>
> ***Q9: Could the authors expand on the importance of used camera pose priors, especially in the context of scene generation? Have the authors started investigating this dimension of the problem?***
>
> **A**: Enlarging the radius range of camera pose is beneficial for scene generation. It is critical for scene to not be with a degraded geometry (a textured sphere). The scene geometry will improve a lot if more advanced camera pose priors are proposed, and we leave it for future work.
>
> ***Q10: Are different text prompts used for different camera poses?***
>
> **A**: Yes, we use view-dependent prompts for pretrained Stable Diffusion, following previous works such as Dreamfusion and Magic3D. However, we do not use view-dependent prompts for the variational model (i.e., the LoRA model) since the camera pose has been injected into the model explicitly.
>
> Once again, thank you for your constructive feedback and for considering our paper for acceptance. We'll revise our paper according to your suggestions. We kindly request that you consider raising the score accordingly if we addressed your concerns.

---

> > ### Comment · Reviewer_krEH · 2023-08-14
> >
> > I thank the authors for the detailed and informative rebuttal. I have no additional questions at this point. Thanks a lot!

---

> > > ### Author Response · Authors · 2023-08-17
> > > **Thank you for your feedback**
> > >
> > > We are happy to hear that you find our response satisfactory and are positive on the rating. We will definitely further revise in the final version as promised. Thank you again for the great efforts on reviewing our manuscript and providing the valuable comments.

---

### Official Review · Reviewer_wz2D · 2023-07-03

**Soundness:** 3 good
**Presentation:** 4 excellent
**Contribution:** 4 excellent
**Rating:** 6
**Confidence:** 3

**Summary:**

This paper proposes Variational Score Distillation (VSD), which is a new generalization of Score Distillation Sampling (SDS) that is based on

**Strengths:**

The biggest strength of this paper is likely in its practical applicability; notably, I find it impressive that the paper is able to produce good results with a batch size = 1. Many prior works need larger batch sizes, and especially mesh generation (as in Magic3D) suffer if they do not have large batch sizes which significantly hurts performance. This is significant, as it makes text-to-3D generation much more feasible without access to a large GPU cluster / DGX sort of machine. However, it's not unclear if this is really due to VSD (more on this in the weakness section).

There are also other practical benefits to this algorithm, like the additional diversity in the output that they can offer. This is also significant, as one of the practical needs in deploying a text-to-X system is being able to 'reroll' for better outputs that better align with the user's intent through the text prompt. This is something that VSD uniquely offers.

The paper is also very well written, and the writing is very clear despite the complexity of what is being described. I do have a couple of more detail-oriented questions that I will ask in the questions section.

**Weaknesses:**

I am slightly unconvinced about the effectiveness of VSD when it comes to increased quality.

First, as a baseline, the generated results qualitatively do look much better than DreamFusion or Magic3D. The user studies agree, with an overwhelming margin (90+%).

However, it is also the case that the generated results in this paper _without_ VSD and with just the higher resolution prior (and annealed t) already looks arguably significantly better than DreamFusion or Magic3D. Without a user study to then compare against this result with and without VSD, it's hard to make the claim that VSD itself is the factor that contributes most significantly to the improved quality. It's not entirely made clear in the paper what exactly is the factor that leads to this significantly quality increase. Empirically, high resolution NeRF training for SDS with Stable Diffusion has been difficult to make work (just even based on open source re-implementations of various papers), _especially_ with a small batch size of N=1 that they claim in the paper... so I am very curious what leads to the quality increase. But also it makes me question whether VSD really adds much in terms of quality without more examples or a user study. (Figure 5 does show an example where there seems to be perceived difference, but in my own experience, this is a minor difference that could just be caused by a slightly different hyper parameter)

Figure 17, although it's only on a single example, somewhat confirms this since the high CFG weight seems to result in very similar results with the VSD results with low CFG. Although high CFG weight is a factor for over saturation, some of the results with VSD also seems to suffer from over saturation anyways.

Regardless, VSD does offer other benefits like diversity. But it would be amazing to hear from the authors on their thoughts on this quality difference in the SDS baseline. Ideally a user study can compare between their baseline NeRF generation results with SDS against VSD, but I understand if that's difficult.

At the very least, there should be more examples (beyond the single example they show in Figure 5) that compare between their baseline NeRF results with SDS vs VSD. This would make this paper an extremely solid contribution. (it's also possible I just missed something).

**Questions:**

1. In Figure 3, the VSD results generate slightly more 'oversaturated' results in comparison to ancestral sampling. Similarly, the mesh results in the paper generally seem to still suffer from over saturation issues (although some of this is likely just due to not modeling accurate physically based rendering). Is there any explanation for the perceived differences between VSD and ancestral sampling and why they lead to these qualitative differences?

2. For the SDS experiments (in Figure 17 in the appendix, for example), what are the experimental settings? Do they use the same batch size = 1 with all of the same hyper parameters as the VSD? What hyper parameter differences exist between this and DreamFusion, Magic3D?

3. How does the SDS NeRF results from this paper (without VSD) compare to DreamFusion and Magic3D in terms of perceived quality? What are the factors that cause this quality difference?

**Limitations:**

The limitations are addressed, although I have a lot of questions about the actual efficacy of VSD with respect to quality improvements. It would also be nice to have a broader statement on the societal impacts (even if it's somewhere in the appendix, unless I just didn't see it) since text-to-3D models are something that can bring potentially harmful impacts to the labor market around content creation.

---

> ### Author Rebuttal · Authors · 2023-08-10
>
> Thank you for your valuable review and suggestions. Below we provide a point-to-point response to all comments. If our response has addressed the concerns and brings new insights to the reviewer, we will highly appreciate it if the reviewer considers raising the score.
>
> ***Q1: What exactly is the factor that leads to this significantly quality increase? Directly compare VSD with SDS? Oversaturated in Figure 17?***
>
> **A**: Thanks for the constructive comments. We believe that VSD itself is the main factor to improve quality and add both quantiative and qualitative results to confirm it.
>
> Quantitatively, we fairly compare VSD and SDS in terms of the widely adopted FID score in both 2D and 3D experiments (see details in ***common response Q1***). In particular, we compare VSD with SDS in the same setting of 512 resolution and annealed $t$ in the 3D experiments to eliminate the effects of other factors. Qualitatively, Figure 17 shows that VSD relieves the oversatuated problem of SDS. We provide extra results comparing VSD and SDS in ***response pdf, Fig.1***, including NeRF training stage-1 and mesh texturing stage-3, both with 512 resolution and annealed $t$.
>
> In all experiments, VSD itself significantly outperforms SDS, showing its effectiveness. We will add all new results in the final version and we believe that the quality of the paper will be significantly improved.
>
>
> ***Q2: VSD is slightly more oversaturated than ancestral sampling. Is there any explanation for the perceived differences between VSD and ancestral sampling and why they lead to these qualitative differences?***
>
> **A**: Theoretically, if the variational distributions are sufficiently powerful and the optimization of the variatioinal inference problem is sufficient, then the distribution of samples by VSD will be the same as that by ancestral sampling. However, in practice, we adopt a LoRA model as the variational distribution and optimize it by Adam, which slightly viodate the above conditons and results in the qualitative differences.
>
> Nevertheless, we argue that VSD is still prefarable in zero-shot 3D generation because it significantly outperforms SDS (see more details in response to your Q1) and ancestral sampling can be only used in 2D.
>
>
> ***Q3: What are the experimental settings for the SDS experiments in Fig.17?***
>
> **A**: We use the same settings as VSD (i.e., same hyperparameters, 512 resolution, annealed $t$). In particular, for the 3D representations, we follow most of the experiment settings and hyperparameters in Magic3D, and the differences are: 1. We use 512 resolution and annealed $t$; 2. We use batch size = 1 (following the re-implementation of stable-dreamfusion). 3. We do not use the shading proposed in DreamFusion because we currently find that the generated texture is better without shading. We will make the detailed settings clearer in the final version, and leave adding shading to ProlificDreamer as future work.
>
> ***Q4: How does the SDS NeRF results from this paper (without VSD) compare to DreamFusion and Magic3D in terms of perceived quality? What are the factors that cause this quality difference?***
>
> **A**: Since DreamFusion and Magic3D use a powerful text-to-image base models (Imagen and eDiff-I) than stable-diffusion and not open-sourced, we use the open-sourced re-implementation of them by stable-dreamfusion, and adopt the improvement in the above response of Q3, finding that it improves the performance of SDS in stable-dreamfusion (see Fig.5 in the main text). However, it is still hard to say which is better between our SDS implementation and the official Dreamfusion and Magic3D. Nevertheless, after integrating VSD and the whole pipeline in ProlificDreamer, our generation quality is much better (see user study in appendix K and examples in appendix L) than Dreamfusion and Magic3D.
>
> ***Q5: Add a broader statement on the societal impacts?***
>
> **A**: We have included a broader statement in Sec.6. We will follow the suggestion and add a longer version in the appendix.
>
> Once again, thank you for your constructive feedback and for considering our paper for acceptance. We'll revise our paper according to your suggestions. We kindly request that you consider raising the score accordingly if we addressed your concerns.

---

### Official Review · Reviewer_W2L2 · 2023-07-03

**Soundness:** 3 good
**Presentation:** 3 good
**Contribution:** 3 good
**Rating:** 7
**Confidence:** 4

**Summary:**

The authors introduce the VSD algorithm that generalizes SDS to optimizing a distribution of shapes in terms of KL divergence to the image diffusion model.
Lacking analytic score of the implicit rendered image distribution, they train a surrogate by using LORA finetuning to the base image diffusion model.
The resulting algorithm is demonstrated for optimizing an empirical distribution of particles and interpreted in framework of particle VI.
They empirically find that a lower CFG is possible when using VSD as opposed to SDS which alleviates artefacts (even in the n=1 case).

Additionally, a simple NERF initialization for scenes, annealed time schedule and higher resolution rendering are proposed as orthogonal improvements.

**Strengths:**

Novel method: Using an auxiliary fine-tuned image diffusion model to approximate the score of an implicit distribution is a very neat idea and could have potential wider applicability in other areas of generative modelling.

Presentation: The motivation behind the method, technical background and mathematical framework are presented well and the writing is clear in these aspects. Details regarding the method are also fairly clear.

Strong qualitative result: The ability to use lower CFG is an interesting finding and the authors give evidence for its effectiveness in producing high quality results. Combined with the other practical additions to the training pipeline, this results in very impressive SOTA visual quality.


**Weaknesses:**

Clearer experiments: There are a lot of different experimental settings which dilutes the evaluation of the core VSD contribution imo.
The paper seems to have 4 main experimental settings:
a. Single particle textured mesh objects
b. Single particle NERFs (object and scene)
c. Multi-particle NERF
d. 2D VSD results

In terms of multiple particles, these are only demonstrated for 1 2D prompt and 2 NERF prompts (and maybe 1 mesh in the appendix). Can multiple particle results be demonstrated for the prompts used in a/b.

Given the lack of quantitative metrics, the contributions of VSD, annealing t and higher resolution are only evaluated qualitatively on 1 or 2 prompts. Ablating VSD for more prompts would help strengthen the case for it. Currently it is only done for one of the prompts in setting a.

Lack of quantitative metrics: I understand there aren’t great established metrics in this area but currently the comparison is only done by showing a few qualitative examples and a user study with 5 prompts picked per baseline.
Especially since one of the claims in the paper is that VSD allows diverse generation, quantifying this over a representative set of prompts would greatly strengthen the evaluation section.
R-precision has been reported in previous papers and given the motivation of matching in KL, a conditional FID to the image diffusion model could also be considered.
If computation is a bottleneck, then these could have also been obtained for a coarser version/2D case.

Clarity of explaining VSD results: Although VSD was motivated from the stand point of enabling diverse generations, the empirical benefits it brings in the single particle case is surprising and remains not well explained.
Hence, some of the discussions in section 3.3 (about “superior generalization” and regarding CFG on lines 185,186) seem too vague.
I would recommend the authors could change the wording to make it more clear which results are empirical findings that may not have theoretical justification yet and further to write any intuition/speculation about the reason more clearly.

**Questions:**

- On line 247 it is mentioned that for mesh finetuning, VSD was not helpful for geometry. To clarify, the pipeline for the textured mesh then is: 1. VSD for Nerf init, 2. SDS on DMTet geom finetuning, 3. VSD on Texture finetuning? How would this compare to following Fantasia3D and skipping the 1st NERF stage? Can you show the intermediate results during each stage? Does the SDS stage affect the geometry much in stage 2?
- Is the ice cream ablation in Figure 12 using the textured mesh pipeline?
- In the appendix, there was some intuition about VSD proving more fine/sharp update direction. What do samples from the LoRA tuned model and the corresponding update direction look like?
- The paper claims that SDS is one of the main bottlenecks to scaling resolutions to 512 (line 226). Could you elaborate on this, since it seems they are orthogonal?
- For the 2D experiment, you tried the small U-Net with 2048 particles. Did you also try n=2048 for the LoRA? Are there issues with scaling LoRA?


**Limitations:**

The limitations as expanded upon in the appendix is sufficient. Maybe the authors can expand more about the robustness of the method and sensitivity to hyperparameters which is a common issue in related works.

---

> ### Author Rebuttal · Authors · 2023-08-10
>
> Thank you for your supportive review and suggestions. Below we provide a point-to-point response to all comments. We hope you will find our response satisfactory and raise your score accordingly.
>
>
> ***Q1: Can multiple particle results be demonstrated for the prompts used in a/b？***
>
> **A**: Yes. We provide more multi-particle results in the ***response pdf, Fig.5***, and will add more multi-particle results in the final version.
>
> ***Q2: Comparing VSD and SDS for more prompts.***
>
> **A**: We provide more qualitative results comparing VSD with SDS in the ***response pdf, Fig.1***. We also add quantitative results in ***common response Q1***, including both 2D experiments (evaluated on 1000 prompts) and 3D experiments (evaluated on 100 prompts). In all experiments, VSD significantly outperforms SDS under the same setting, showing the effectiveness of VSD itself.
>
> ***Q3: Lack of quantitative metrics.***
>
> **A**: Quantatively evaluating the results of text-to-3D is challenging. There is no well-estabilished method for this. Since FID score is the most common evaluation metric in 2D generative models, which evaluates both the quality and diversity of the samples, we add quantitative results of FID scores (see details in ***common response Q1***), including both 2D (1000 prompts) and 3D (100 prompts) experiments. In particular, FID scores involve the variance of samples, and thus the better FID scores of VSD suggest that VSD improves the diversity.
>
>
> ***Q4: Improve the writing clarity of explaining VSD results.***
>
> **A**: We will clarify the discussions in Sec.3.3 about the single-particle VSD cases, following the suggestions. In particular, we will clarify the discussions about the empirical findings and theoretical justifications separately, and highlight the intuition of the reason for the benifits by VSD with single particle.
>
> ***Q5: Pipeline and mesh finetuning intermediate results? Does SDS affect much in stage2?***
>
> **A**: Yes, the whole pipeline is: 1. VSD for NeRF, 2. SDS on geom finetuning, 3. VSD on texture finetuning. We show the intermediate results during each stage in the ***response pdf, Fig.3***. We find that converting NeRF to mesh causes lots of detail lost, and using both SDS and VSD can improve the geom results (we provide a detailed explanation for why using SDS in geom stage in ***common response Q2***) and using VSD can further improve the mesh texture.
>
> ***Q6: Compare to Fantasia3D if skipping the 1st NeRF stage?***
>
> **A**: The 1st NeRF stage is crucial for the final results because it does not need any handcrafted initialization for each prompt (as in Fantasia3D), and using VSD in 1st stage can greatly improve both the geometry and texture quality used for initializing stage2 and stage3, since the NeRF quality by VSD is superior to SDS. So we adopt the 1st stage for better performance.
>
>
> Given the NeRF initialization, as shown in ***response pdf, Fig.1***, VSD outperforms SDS significantly in texture optimization. Based on such results, even if skipping the 1st NeRF stage and using a handcrafted initialization as in Fantasia3D, ProlificDreamer still employs VSD to optimize the mesh texture in stage3, which is supposed to obtain a much better texture than SDS (as adopted in Fantasia3D).
>
> ***Q7: Is the ice cream ablation in Figure 12 using the textured mesh pipeline?***
>
> **A**: No. It's the NeRF result from stage-1.
>
> ***Q8: Visualize samples from the LoRA and the corresponding VSD update direction?***
>
> **A**: We provide a sample from the LoRA in ***response pdf, Fig.3***, showing that LoRA samples are consistent to 3D object when optimization converges. Moreover, we visualize VSD/SDS training phase of 2D in ***response pdf, Fig.2***. Since the gradient is not directly readable, we visualize $x+\Delta x$, which is the updated results if current sample optimizes via this gradient direction. As shown in Fig.2, SDS tends to provide over-saturated and over-smooth gradient while VSD provides more natural-looking gradients with more details. As a consequence, VSD provides better final results.
>
> ***Q9: Elaborate why SDS is one of the main bottlenecks to scaling resolutions to 512?***
>
> **A**: We apologize for the potential misunderstanding for the unclear statement. We clarify that SDS is one of the main bottlenecks to generate high-fidelity NeRF, which is orthogonal to the resolution, and the quality by SDS is still poor if simply scaling resolutions to 512. Instead, VSD can provide more details. We provide more examples in ***response pdf, Fig.1.(1)*** to show that 512-res SDS is still much worse than 512-res VSD. We will make it clearer in final version.
>
> ***Q10: Did you also try n=2048 for the LoRA in 2D? Are there issues with scaling LoRA?***
>
> **A**: Following the suggestion, we conduct a new experiment to scale up LoRA in 2D experiments, and find that LoRA obtains better visual quality than U-Net with a large number of particles (e.g. n=2048), so it does not have scaling issues. We will add these results in the final version.
>
> Once again, thank you for your constructive feedback and for considering our paper for acceptance. We'll revise our paper according to your suggestions. We kindly request that you consider raising the score accordingly if we addressed your concerns.

---

> > ### Comment · Reviewer_W2L2 · 2023-08-14
> >
> > I thank the authors for the detailed response. I have read the rebuttal in full.
> > The extra quantitative results using FID help strengthen the case for VSD a lot and I expect these to be incorporated in the final paper.
> >
> > One additional thing regarding the evaluation of multiple particle VSD:
> > “For VSD(n=4), as we can get 4 particles (3D objects) per prompt, we randomly select one particle per prompt and render the corresponding 10 images of the selected particle for fair comparison”
> > I agree this is definitely a way to make comparison SDS very “fair”. But if a single model is selected per prompt, this would not highlight the additional diversity from VSD, so it might be unfair to VSD.
> >
> > You could instead do for say 16 views, you select 4 from the 1st particle, 4 from the 2nd particle, etc… This would allow you to have renderings from all particles in your FID evaluation without changing the number of input views. It may help strengthen the evaluation for the VSD (and highlight the ability for it to get diversity).
> >
> > This addresses my concern about the quantitative metrics and my other questions, so I will raise my score to accept.

---

> > > ### Author Response · Authors · 2023-08-16
> > > **Thank you for your feedback**
> > >
> > > Thank you for the detailed feedback and for raising the score, and thank you for providing the valuable suggestion for evaluating multi-particle VSD. We follow your suggestion and re-evaluate VSD(n=4) by considering all the particles and find that the FID by VSD(n=4) is significantly better than SDS and VSD(n=1). The detailed results are:
> > >
> > > | Method(3D) | SDS    | VSD(n=1) | VSD(n=4) |
> > > | :--------- | :----- | :------- | :------- |
> > > | 3D-FID(↓)  | 191.82 | 186.87   | 173.96   |
> > >
> > > Specifically, for each prompt, we can get 4 particles by VSD(n=4). For each particle, we render 10 images by 10 views, following the same setting specified before, and then get 40 images per prompt. Then we randomly select 10 images per prompt and get the same number of images as the baseline settings, and then compute the FID score correspondingly. We repeat such process for 10 times and compute the mean FID score, as shown above.
> > >
> > > We appreciate your detailed comments and suggestions. Thank you again for your great effort!

---

### Author Rebuttal · Authors · 2023-08-10

We sincerely thank all reviewers's efforts and their appreciation of our novel contributions as well as very detailed and insightful suggestions to further improve our paper. We find there are common concerns to our paper, and we'd like to clarify here. We also add a pdf file to add more experiment results.

***Q1: Quantitative evaluation for VSD and SDS? (from W2L2,wz2D,krEH)***
- Table a: 3D Sample Quality by SDS or VSD, 100 prompts.

|Method(3D)|SDS|VSD(n=1)|
|-|-|-|
|3D-FID(↓)|118.92|107.02|

- Table b: 3D Sample Quality by SDS or VSD, 25 prompts.

|Method(3D)|SDS|VSD(n=1)|VSD(n=4)|
|-|-|-|-|
|3D-FID(↓)|191.82|186.87|185.88|

- Table c: 2D Sample Quality by Different Samplers, 1000 prompts.

|Method(2D)|SDS|VSD(n=4)|VSD(n=8)|DPM++|
|-|-|-|-|-|
|FID(↓)|90.09|68.02|66.68|47.91|



**A**: We provide quantitative results (FID score, following common 2D generative model evaluations) for comparing VSD and SDS in both 2D and 3D experiments, as shown in Table a, Table b and Table c. We find that **VSD outperforms SDS** in both 2D and 3D cases, which confirms our claim about the significance of VSD. We present the detailed settings and results as follows.

**Detailed settings**:
- For 3D experiments, we compute the FID score between rendered images by SDS/VSD and 2D sampled images by ancestral sampling, named as 3D-FID. Specifically, we select 100 prompts from previous works including DreamFusion, Magic3D and Fantasia3D. For each prompt, we use VSD (#particles n=1, CFG=7.5) or SDS (CFG=100) to optimize one 3D object and render 10 images uniformly from the circumference at an angle of 30° above the horizon, and collect 1k images in total. To isolatedly compare VSD with SDS, we run with the default setting of the stage-1 NeRF training of ProlificDreamer (i.e., both VSD and SDS are in 512 resolution and use annealed $t$).
    - In Table a, We compute the FID score between the 1k samples from the 100 prompts and a 50k reference batch, which is sampled by 50-step DPM-Solver++ with 500 images per prompt.
    - In Table b, due to the time and computation resource limits, we compare the results for VSD with n=4 under only 25 randomly-selected prompts from the aforementioned 100 prompts, and compare SDS, VSD(n=1) and VSD(n=4) with the 50-step DPM-Solver++ under these 25 prompts. For VSD(n=4), as we can get 4 particles (3D objects) per prompt, we randomly select one particle per prompt and render the corresponding 10 images of the selected particle for fair comparison.

- For 2D experiments in Table c, we follow the common setting of evaluating text-to-image models by computing FID on MSCOCO2014 validation set. Specifically, we randomly select 1k prompts and sample one image per prompt by either 50-step DPM-Solver++, SDS(CFG=100), VSD(n=4,CFG=7.5) or VSD(n=8,CFG=7.5) to collect 1k samples for each method, and then compute the FID between the obtained samples and the entire COCO validation set. For VSD, as we can get n images per prompt, we randomly select one image per prompt for fair comparison.

**Results**:
- **VSD with n=1 still outperforms SDS in 3D (both with 512 resolution and annealed $t$)**, as shown in Table a and Table b. Such quantitative results agree with the quanlitative results shown in ***response pdf, Fig.1***, showing the effectiveness of VSD.
- **Using more particles is slightly better**. Due to the limitation of time and computation resoures, we only compare n=1 and n=4 in 3D experiments, and n=4 with n=8 in the 2D experiments. As shown in Table b, VSD with 4 particles slightly outperforms VSD with 1 particles in the 3D setting; and as shown in Table c, VSD with 8 particles slightly outperforms VSD with 4 particles in the 2D setting.
- **VSD outperforms SDS in 2D**. As shown in Table c, the FID by VSD is much better than SDS. As the 2D setting isolates the sampling algorithm from the 3D representations, we can directly compare different sampling algorithms, finding that VSD can get better sample quality than SDS (though still worse than SOTA diffusion samplers, but it can generalize to 3D cases).


***Q2: Why using SDS in stage-2 for the geometry optimization of mesh? (from W2L2,krEH,eq7N)***

**A**: VSD can also be used to generate geometry. To validate this, we provide an ablation example in the ***response pdf, Fig.(3a),(3b)***. As shown in the figure, VSD can obtain reasonable geometry. Although the some part of the geometry from VSD is with more details than SDS (including the tail of the horse), on the whole, the result from VSD is similar with SDS. We conjecture that this is because currently the triangle size of the mesh is relatively large and can't represent very fine details. Thus, for efficiency, we use SDS instead of VSD for mesh geometry optimization. We believe that VSD can be used to obtain high quality mesh if more advanced mesh represetation is available.

Moreover, despite that we use SDS to optmize the geometry in stage-2, VSD is still crucial in stage-1 and stage-3, in which VSD significantly improves the generated quality.

***Q3: How will the number of particles affect the results of VSD? (from krEH,Gck3,eq7N)***
**A**: In general, using more particles will make the variational distribution (LoRA model) easier to train, and thus improve the sample quality. We provide an empirical evidence in Table b and
c (see more details in the response of Q1). Moreover, as LoRA has a strong image prior and is suitable for few-shot learning, we find that using 1~4 particles is enough for high-quality 3D generation.


We think the the quality of our work has been improved a lot. We are welcome for further questions.

---

### Decision · Program_Chairs · 2023-09-21

**Decision:**

Accept (spotlight)

**Comment:**

The paper proposed a novel theoretic perspective for text to 3D generation, and introduced variational score distillation which optimizes the KL divergence between image diffusion and the distribution of 3D shapes. The surrogate is implemented as LORA fine-tuning. Qualitative results demonstrated the effectiveness of VSD on the diversity of generated 3D assets, the over-saturation and the over-smoothing problem. The additional quantitative results from the rebuttal further strengthen the paper, and all the reviewers are positive about the paper. The proposed VSD poses a new avenue for a theoretical understanding of text-to-3D generation, can be applied to many other areas of 3D generative modelling, and can also inspire follow-ups on the related directions, and thus I recommend accepting the paper.

As promised during the rebuttal and discussions, please revise the paper by adding the provided quantitative evaluations (already presented in the rebuttal), adding more clarifications on the experimental settings (whether or not using SDS on stage-2 training), more qualitative results of multi-particle experiments, and moving related work to the main paper, etc.